# ON STOCHASTIC SIGN DESCENT METHODS

## ABSTRACT

Various gradient compression schemes have been proposed to mitigate the communication cost in distributed training of large scale machine learning models. Sign-based methods, such as signSGD (Bernstein et al., 2018), have recently been gaining popularity because of their simple compression rule and connection to adaptive gradient methods, like ADAM. In this paper, we perform a general analysis of sign-based methods for non-convex optimization. Our analysis is built on intuitive bounds on success probabilities and does not rely on special noise distributions nor on the boundedness of the variance of stochastic gradients. Extending the theory to distributed setting within a parameter server framework, we assure exponentially fast variance reduction with respect to number of nodes, maintaining 1-bit compression in both directions and using small mini-batch sizes. We validate our theoretical findings experimentally.

## 1 INTRODUCTION

One of the key factors behind the success of modern machine learning models is the availability of large amounts of training data (Bottou & Le Cun, 2003; Krizhevsky et al., 2012; Schmidhuber, 2015). However, the state-of-the-art deep learning models deployed in industry typically rely on datasets too large to fit the memory of a single computer, and hence the training data is typically split and stored across a number of compute nodes capable of working in parallel. Training such models then amounts to solving optimization problems of the form

$$\min_{x \in \mathbb{R}^d} f(x) \coloneqq \frac{1}{M} \sum_{m=1}^{M} f_m(x), \tag{1}$$

where $f_m : \mathbb{R}^d \to \mathbb{R}$ represents the *non-convex* loss of a deep learning model parameterized by $x \in \mathbb{R}^d$ associated with data stored on node $m$.

Arguably, stochastic gradient descent (SGD) (Robbins & Monro, 1951; Vaswani et al., 2019; Qian et al., 2019) in of its many variants (Kingma & Ba, 2015; Duchi et al., 2011; Schmidt et al., 2017; Zeiler, 2012; Ghadimi & Lan, 2013) is the most popular algorithm for solving (1). In its basic implementation, all workers $m \in \{1, 2, \ldots, M\}$ in parallel compute a random approximation $\hat{g}^m(x_k)$ of $\nabla f_m(x_k)$, known as the *stochastic gradient*. These approximations are then sent to a master node which performs the aggregation

$$\hat{g}(x_k) \coloneqq \frac{1}{M} \sum_{m=1}^{M} \hat{g}^m(x_k).$$

The aggregated vector is subsequently broadcast back to the nodes, each of which performs an update of the form

$$x_{k+1} = x_k - \gamma_k \hat{g}(x_k),$$

thus updating their local copies of the parameters of the model.

### 1.1 GRADIENT COMPRESSION

Typically, communication of the local gradient estimators $\hat{g}^m(x_k)$ to the master forms the bottleneck of such a system (Seide et al., 2014; Zhang et al., 2017; Lin et al., 2018). In an attempt to alleviate this communication bottleneck, a number of compression schemes for gradient updates have been proposed and analyzed (Alistarh et al., 2017; Wang et al., 2018; Wen et al., 2017; Khirirat et al., 2018;

Table 1: Summary of the theoretical results obtained in this work. $\tilde{\mathcal{O}}$ notation ignores logarithmic factors and $\mathcal{O}^*$ notation shows the rate to a neighbourhood of the solution.

| | This work
Theorem 1 | This work
Theorem 2 | SGD (Sec. C)
Theorem 6 | (Bernstein et al., 2019)
signSGD, Theorem 1 |
|---|---|---|---|---|
| Step size
$\gamma_k \equiv \gamma$ | $\mathcal{O}^*\left(\frac{1}{K}\right)$ | $\mathcal{O}^*\left(\frac{1}{K}\right)$ | $\mathcal{O}^*\left(\frac{1}{K}\right)$ | ✗ |
| Step size
$\gamma_k = \frac{\gamma_0}{\sqrt{k+1}}$ | $\tilde{\mathcal{O}}\left(\frac{1}{\sqrt{K}}\right)$ | $\mathcal{O}\left(\frac{1}{\sqrt{K}}\right)$ | $\tilde{\mathcal{O}}\left(\frac{1}{\sqrt{K}}\right)$ | ✗ |
| Step size
$\gamma = \mathcal{O}\left(\frac{1}{\sqrt{K}}\right)$ | $\mathcal{O}\left(\frac{1}{\sqrt{K}}\right)$ | $\mathcal{O}\left(\frac{1}{\sqrt{K}}\right)$ | $\mathcal{O}\left(\frac{1}{\sqrt{K}}\right)$ | $\mathcal{O}\left(\frac{1}{\sqrt{K}}\right)$ |
| Can handle biased
estimators? | ✓ | ✓ | ✗ | ✗ |
| Weak dependence on
smoothness parameters? | ✓ $\frac{1}{d}\sum_{i=1}^{d} L_i$ | ✓ $\frac{1}{d}\sum_{i=1}^{d} L_i$ | ✗ $\max_{i}^{d} L_i$ | ✓ $\frac{1}{d}\sum_{i=1}^{d} L_i$ |
| Weak noise
assumptions? | ✓
$\rho_i > \frac{1}{2}$ | ✓
$\rho_i > \frac{1}{2}$ | ✗
$\mathbb{E}\|\hat{g}\|_2^2 \leq C$ [1] | ✗
unimodal,
symmetric &
$\text{Var}[\hat{g}_i] \leq \sigma_i^2$ |
| Gradient norm used
in theory | $\rho$-norm | $\rho$-norm | (squared) $l^2$ | A mix of
$l^1$ and $l^2$ norms |

Mishchenko et al., 2019). A *compression scheme* is a (possibly randomized) mapping $Q : \mathbb{R}^d \to \mathbb{R}^d$, applied by the nodes to $\hat{g}^m(x_k)$ (and possibly also by the master to aggregated update in situations when broadcasting is expensive as well) in order to reduce the number of bits of the communicated message.

**Sign-based compression.** Although most of the existing theory is limited to *unbiased* compression schemes, i.e., on operators $Q$ satisfying $\mathbb{E}Q(x) = x$, *biased* schemes such as those based on communicating signs of the update entries only often perform much better (Seide et al., 2014; Strom, 2015; Wen et al., 2017; Carlson et al., 2015; Balles & Hennig, 2018; Bernstein et al., 2018; 2019; Zaheer et al., 2018; Liu et al., 2019). The simplest among these sign-based methods is signSGD (see also Algorithm 1; Option 1), whose update direction is assembled from the component-wise signs of the stochastic gradient.

**Adaptive methods.** While ADAM is one of the most popular *adaptive* optimization methods used in deep learning (Kingma & Ba, 2015), there are issues with its convergence (Reddi et al., 2019) and generalization (Wilson et al., 2017) properties. It was noted in Balles & Hennig (2018) that the behaviour of ADAM is similar to a momentum version of signSGD. Connection between sign-based and adaptive methods has long history, originating at least in Rprop (Riedmiller & Braun, 1993) and RMSprop (Tieleman & Hinton, 2012). Therefore, investigating the behavior of signSGD can improve our understanding on the convergence of adaptive methods such as ADAM.

## 1.2 CONTRIBUTIONS

We now summarize the main contributions of this work. Our key results are summarized in Table 1.

---

[1] In fact, bounded variance assumption, being weaker than bounded second moment assumption, is stronger (or, to be strict, more curtain) than SPB assumption in the sense of differential entropy, but not in the direct sense. The entropy of probability distribution under the bounded variance assumption is bounded, while under the SPB assumption it could be arbitrarily large. This observation is followed by the fact that for continuous random variables, the Gaussian distribution has the maximum differential entropy for a given variance (see `https://en.wikipedia.org/wiki/Differential_entropy`).

- **2 methods for 1-node setup.** In the $M = 1$ case, we study two general classes of sign based methods for minimizing a smooth non-convex function $f$. The first method has the standard form[2]

$$x_{k+1} \leftarrow x_k - \gamma_k \operatorname{sign} \hat{g}(x_k), \tag{2}$$

while the second has a new form not considered in the literature before:

$$x_{k+1} \leftarrow \arg\min\{f(x_k), f(x_k - \gamma_k \operatorname{sign} \hat{g}(x_k))\}. \tag{3}$$

- **Key novelty.** The key novelty of our methods is in a *substantial relaxation* of the requirements that need to be imposed on the gradient estimator $\hat{g}(x_k)$ of the true gradient $\nabla f(x^k)$. In sharp contrast with existing approaches, we allow $\hat{g}(x_k)$ to be *biased*. Remarkably, we only need one additional and rather weak assumption on $\hat{g}(x_k)$ for the methods to provably converge: we require the signs of the entries of $\hat{g}(x_k)$ to be equal to the signs of the entries of $\nabla f(x^k)$ with a probability strictly larger than $1/2$ (see Section 2; Assumption 1). We show through a counterexample (see Section 2.2) that this assumption is *necessary*.

- **Geometry.** As a byproduct of our analysis, we uncover a *mixed $l^1$-$l^2$ geometry* of sign descent methods (see Section 3).

- **Convergence theory.** We perform a complexity analysis of methods (2) and (3) (see Section 4.1; Theorem 1). While our complexity bounds have the same $\mathcal{O}(1/\sqrt{K})$ dependence on the number of iterations, they have a *better dependence on the smoothness parameters* associated with $f$. Theorem 1 is the first result on signSGD for non-convex functions which does not rely on mini-batching, and which allows for step sizes independent of the total number of iterations $K$. Finally, Theorem 1 in Bernstein et al. (2019) can be recovered from our general Theorem 1. Our bounds are cast in terms of a *novel norm-like function, which we call the $\rho$-norm*, which is a weighted $l^1$ norm with positive variable weights.

- **Distributed setup.** We extend our results to the *distributed setting* with arbitrary $M$ (Section 4.2), where we also consider sign-based compression of the aggregated gradients.

## 2 SUCCESS PROBABILITIES AND GRADIENT NOISE

In this section we describe our key (and weak) assumption on the gradient estimator $\hat{g}(x)$ of the true gradient $\nabla f(x)$, and give an example which shows that without this assumption, method (2) can fail.

### 2.1 SUCCESS PROBABILITY BOUNDS

**Assumption 1** (SPB: Success Probability Bounds). *For any $x \in \mathbb{R}^d$, we have access to an independent (and not necessarily unbiased) estimator $\hat{g}(x)$ of the true gradient $g(x) \coloneqq \nabla f(x)$ that satisfies*

$$\rho_i(x) \coloneqq \operatorname{Prob}\left(\operatorname{sign} \hat{g}_i(x) = \operatorname{sign} g_i(x)\right) > \tfrac{1}{2}, \quad if \quad g_i(x) \neq 0 \tag{4}$$

*for all $x \in \mathbb{R}^d$ and all $i \in \{1, 2, \ldots, d\}$.*

We will refer to the probabilities $\rho_i$ as *success probabilities*. As we will see, they play a central role in the convergence of sign based methods. We stress that *Assumption 1* is the *only* assumption on gradient noise in this paper. Moreover, we argue that it is reasonable to require from the sign of stochastic gradient to show true gradient direction more likely than the opposite one. Extreme cases of this assumption are the absence of gradient noise, in which case $\rho_i = 1$, and an overly noisy stochastic gradient, in which case $\rho_i \approx \frac{1}{2}$.

**Remark 1.** *Assumption 1 can be relaxed by replacing bounds (4) with*

$$\mathbb{E}\left[\operatorname{sign}\left(\hat{g}_i(x) \cdot g_i(x)\right)\right] > 0, \quad if \quad g_i(x) \neq 0.$$

*However, if $\operatorname{Prob}(\operatorname{sign} \hat{g}_i(x) = 0) = 0$ (e.g. in the case of $\hat{g}_i(x)$ has continuous distributions), then these two bounds are identical.*

---

[2] sign $g$ is applied element-wise to the entries $g_1, g_2, \ldots, g_d$ of $g \in \mathbb{R}^d$. For $t \in \mathbb{R}$ we define sign $t = 1$ if $t > 0$, sign $t = 0$ if $t = 0$, and sign $t = -1$ if $t < 0$.

**Extension to stochastic sign oracle.** Notice that we do *not* require $\hat{g}$ to be unbiased. Moreover, we do *not* assume uniform boundedness of the variance, or of the second moment. This observation allows to extend existing theory to more general sign-based methods with a stochastic sign oracle. By a stochastic sign oracle we mean an oracle that takes $x_k \in \mathbb{R}^d$ as an input, and outputs a random vector $\hat{s}_k \in \mathbb{R}^d$ with entries in $\pm 1$. However, for the sake of simplicity, in the rest of the paper we will work with the signSGD formulation, i.e., we let $\hat{s}_k = \operatorname{sign} \hat{g}(x_k)$.

## 2.2 A counterexample to signSGD

Here we analyze a counterexample to signSGD discussed in Karimireddy et al. (2019). Consider the following least-squares problem with unique minimizer $x^* = (0, 0)$:

$$\min_{x \in \mathbb{R}^2} f(x) = \tfrac{1}{2} \left[ \langle a_1, x \rangle^2 + \langle a_2, x \rangle^2 \right], \quad a_1 = \left[ \begin{smallmatrix} 1+\varepsilon \\ -1+\varepsilon \end{smallmatrix} \right], a_2 = \left[ \begin{smallmatrix} -1+\varepsilon \\ 1+\varepsilon \end{smallmatrix} \right],$$

where $\varepsilon \in (0, 1)$ and stochastic gradient $\hat{g}(x) = \nabla \langle a_i, x \rangle^2 = 2 \langle a_i, x \rangle a_i$ with probabilities $1/2$ for $i = 1, 2$. Let us take any point from the line $l = \{(z_1, z_2) : z_1 + z_2 = 2\}$ as initial point $x_0$ for the algorithm and notice that $\operatorname{sign} \hat{g}(x) = \pm(1, -1)$ for any $x \in l$. Therefore, signSGD with any step-size sequence remains stuck along the line $l$, whereas the problem has a unique minimizer at the origin.

We now investigate the cause of the divergence. In this counterexample, Assumption 1 is violated. Indeed, note that

$$\operatorname{sign} \hat{g}(x) = (-1)^i \operatorname{sign} \langle a_i, x \rangle \left[ \begin{smallmatrix} -1 \\ 1 \end{smallmatrix} \right] \quad \text{with probabilities} \quad \tfrac{1}{2} \quad \text{for} \quad i = 1, 2.$$

By $S := \{ x \in \mathbb{R}^2 : \langle a_1, x \rangle \cdot \langle a_2, x \rangle > 0 \} \neq \emptyset$ denote the open cone of points having either an acute or an obtuse angle with both $a_i$'s. Then for any $x \in S$, the sign of the stochastic gradient is $\pm(1, -1)$ with probabilities $1/2$. Hence for any $x \in S$, we have low success probabilities:

$$\rho_i(x) = \operatorname{Prob} \left( \operatorname{sign} \hat{g}_i(x) = \operatorname{sign} g_i(x) \right) \leq \tfrac{1}{2}, \quad i = 1, 2.$$

So, in this case we have an entire conic region with low success probabilities, which clearly violates (4). Furthermore, if we take a point from the complement open cone $\bar{S}^c$, then the sign of stochastic gradient equals to the sign of gradient, which is perpendicular to the axis of $S$ (thus in the next step of the iteration we get closer to $S$). For example, if $\langle a_1, x \rangle < 0$ and $\langle a_2, x \rangle > 0$, then $\operatorname{sign} \hat{g}(x) = (1, -1)$ with probability 1, in which case $x - \gamma \operatorname{sign} \hat{g}(x)$ gets closer to low success probability region $S$.

In summary, in this counterexample there is a conic region where the sign of the stochastic gradient is useless (or behaves adversarially), and for any point outside that region, moving direction (which is the opposite of the sign of gradient) leads toward that conic region.

## 2.3 Sufficient conditions for SPB

To justify our SPB assumption, we show that it holds under general assumptions on gradient noise.

**Lemma 1** (see B.1). *Assume that for any point $x \in \mathbb{R}^d$, we have access to an independent and unbiased estimator $\hat{g}(x)$ of the true gradient $g(x)$. Assume further that each coordinate $\hat{g}_i$ has a unimodal and symmetric distribution with variance $\sigma_i^2 = \sigma_i^2(x)$, $1 \leq i \leq d$. Then $\rho_i \geq \tfrac{1}{2} + \tfrac{1}{2} \frac{|g_i|}{|g_i| + \sqrt{3}\sigma_i} > \tfrac{1}{2}$ if $g_i \neq 0$.*

Next, we remove the distribution condition and add a strong growth condition (Schmidt & Le Roux, 2013; Vaswani et al., 2019) together with fixed mini-batch size.

**Lemma 2** (see B.2). *Assume that for any point $x \in \mathbb{R}^d$, we have access to an independent, unbiased estimator $\hat{g}(x)$ of the true gradient $g(x)$, with coordinate-wise bounded variances $\sigma_i^2(x) \leq c\, g_i^2(x)$ for some constant c. Then, choosing a mini-batch size $\tau > 2c$, we get $\rho_i \geq 1 - c/\tau > \tfrac{1}{2}$, if $g_i \neq 0$.*

Finally, we give an adaptive condition on mini-batch size for the SPB assumption to hold.

**Lemma 3** (see B.3). *Assume that for any point $x \in \mathbb{R}^d$ we have access to an independent and unbiased estimator $\hat{g}(x)$ of the true gradient $g(x)$. Let $\sigma_i^2 = \sigma_i^2(x)$ be the variance and $\nu_i^3 = \nu_i^3(x)$ be the 3th central moment of $\hat{g}_i(x)$, $1 \leq i \leq d$. Then SPB assumption holds if mini-batch size $\tau > 2 \min \left( \sigma_i^2 / g_i^2, \nu_i^3 / |g_i| \sigma_i^2 \right)$.*



Figure 1: Contour plots of the $l^{1,2}$ norm (5) at 4 different scales with fixed noise $\sigma = 1$.

## 3 A NEW "NORM" FOR MEASURING THE SIZE OF THE GRADIENTS

In this section we introduce the concept of a norm-like function, which call $\rho$-*norm*, induced from success probabilities. Used to measure gradients in our convergence rates, $\rho$-norm is a technical tool enabling the analysis.

**Definition 1** ($\rho$-norm). *Let $\rho := \{\rho_i(x)\}_{i=1}^d$ be the collection of probability functions from the SPB assumption. We define the $\rho$-norm of gradient $g(x)$ via $\|g(x)\|_\rho := \sum_{i=1}^d (2\rho_i(x) - 1)|g_i(x)|$.*

Note that $\rho$-norm is not a norm as it may not satisfy the triangle inequality. However, under SPB assumption, $\rho$-norm is positive definite as it is a weighted $l^1$ norm with positive (and variable) weights $2\rho_i(x) - 1 > 0$. That is, $\|g\|_\rho \geq 0$, and $\|g\|_\rho = 0$ if and only if $g = 0$. Under the assumptions of Lemma 2, $\rho$-norm can be lower bounded by a weighted $l^1$ norm with positive constant weights $1 - 2c_i^2 > 0$: $\|g\|_\rho = \sum_{i=1}^d (2\rho_i - 1)|g_i| \geq \sum_{i=1}^d (1 - 2c_i^2)|g_i|$. Under the assumptions of Lemma 1, $\rho$-norm can be lower bounded by a mixture of the $l^1$ and squared $l^2$ norms:

$$\|g\|_\rho = \sum_{i=1}^d (2\rho_i - 1)|g_i| \geq \sum_{i=1}^d \frac{g_i^2}{|g_i| + \sqrt{3}\sigma_i} := \|g\|_{l^{1,2}}. \tag{5}$$

Note that $l^{1,2}$-norm is again not a norm. However, it is positive definite, continuous and order preserving, i.e., for any $g^k$, $g$, $\tilde{g} \in \mathbb{R}^d$ we have: i) $\|g\|_{l^{1,2}} \geq 0$ and $\|g\|_{l^{1,2}} = 0$ if and only if $g = 0$; ii) $g^k \to g$ (in $l^2$ sense) implies $\|g^k\|_{l^{1,2}} \to \|g\|_{l^{1,2}}$, and iii) $0 \leq g_i \leq \tilde{g}_i$ for any $1 \leq i \leq d$ implies $\|g\|_{l^{1,2}} \leq \|\tilde{g}\|_{l^{1,2}}$. From these three properties it follows that $\|g^k\|_{l^{1,2}} \to 0$ implies $g^k \to 0$. These properties are important as we will measure convergence rate in terms of the $l^{1,2}$ norm in the case of unimodal and symmetric noise assumption. To understand the nature of the $l^{1,2}$ norm, consider the following two cases when $\sigma_i(x) \leq c|g_i(x)| + \tilde{c}$ for some constants $c$, $\tilde{c} \geq 0$. If the iterations are in $\varepsilon$-neighbourhood of a minimizer $x^*$ with respect to the $l^\infty$ norm (i.e., $\max_{1 \leq i \leq d} |g_i| \leq \varepsilon$), then the $l^{1,2}$ norm is equivalent to scaled $l^2$ norm squared: $\frac{1}{(1+\sqrt{3}c)\varepsilon + \sqrt{3}\tilde{c}}\|g\|_2^2 \leq \|g\|_{l^{1,2}} \leq \frac{1}{\sqrt{3}\tilde{c}}\|g\|_2^2$. On the other hand, if iterations are away from a minimizer (i.e., $\min_{1 \leq i \leq d} |g_i| \geq L$), then the $l^{1,2}$-norm is equivalent to scaled $l^1$ norm: $\frac{1}{1+\sqrt{3}(c+\tilde{c}/L)}\|g\|_1 \leq \|g\|_{l^{1,2}} \leq \frac{1}{1+\sqrt{3}c}\|g\|_1$. These equivalences are visible in Figure 1, where we plot the level sets of $g \mapsto \|g\|_{l^{1,2}}$ at various distances from the origin. Similar mixed norm observation was also noted in Bernstein et al. (2019).

## 4 CONVERGENCE THEORY

Now we turn to our theoretical results of sign based methods. First we give our general convergence results under the SPB assumption. Afterwards, we present convergence result in the distributed setting under the unimodal and symmetric noise assumptions.

Throughout the paper we assume that $f \colon \mathbb{R}^d \to \mathbb{R}$ is lower bounded, i.e., $f(x) \geq f^*$, $x \in \mathbb{R}^d$ and is $L$-smooth with some non-negative constants $L = [L_1, \ldots, L_d]$. That is, we assume that $f(y) \leq f(x) + \langle \nabla f(x), y - x \rangle + \sum_{i=1}^d \frac{L_i}{2}(y_i - x_i)^2$ for all $x$, $y \in \mathbb{R}^d$. We allow $f$ to be nonconvex. Let $\bar{L} := \frac{1}{d}\sum_i L_i$ and $L_{\max} = \max_i L_i$.

### 4.1 CONVERGENCE ANALYSIS FOR $M = 1$

We now state our convergence result for Algorithm 1 under the general SPB assumption.

---

**Algorithm 1** SIGNSGD

---
1: **Input:** step size $\gamma_k$, current point $x_k$
2: $\hat{g}_k \leftarrow \text{StochasticGradient}(x_k)$
3: **Option 1:** $x_{k+1} \leftarrow x_k - \gamma_k \operatorname{sign} \hat{g}_k$
4: **Option 2:** $x_{k+1} \leftarrow \arg\min\{f(x_k), f(x_k - \gamma_k \operatorname{sign} \hat{g}_k)\}$

---

**Theorem 1** (Non-convex convergence of signSGD, see B.4). *Under the SPB assumption, signSGD (Algorithm 1 with Option 1) with step sizes $\gamma_k = \gamma_0/\sqrt{k+1}$ converges as follows*

$$\min_{0 \le k < K} \mathbb{E}\|\nabla f(x_k)\|_\rho \le \frac{1}{\sqrt{K}}\left[\frac{f(x_0)-f^*}{\gamma_0} + \gamma_0 d\bar{L}\right] + \frac{\gamma_0 d\bar{L}}{2}\frac{\log K}{\sqrt{K}} \ . \tag{6}$$

*If $\gamma_k \equiv \gamma > 0$, we get $1/K$ convergence to a neighbourhood of the solution:*

$$\frac{1}{K}\sum_{k=0}^{K-1} \mathbb{E}\|\nabla f(x_k)\|_\rho \le \frac{f(x_0)-f^*}{\gamma K} + \frac{d\bar{L}}{2}\gamma \ . \tag{7}$$

We now comment on the above result:

• **Generalization.** Theorem 1 is the first general result on signSGD for non-convex functions without mini-batching, and with step sizes independent of the total number of iterations $K$. Known convergence results (Bernstein et al., 2018; 2019) on signSGD use mini-batches and/or step sizes dependent on $K$. Moreover, they also use unbiasedness and unimodal symmetric noise assumptions, which are stronger assumptions than our SPB assumption (see Lemma 1). Finally, Theorem 1 in Bernstein et al. (2019) can be recovered from Theorem 1 (see Section D for the details).

• **Convergence rate.** Rates (6) and (7) can be arbitrarily slow, depending on the probabilities $\rho_i$. This is to be expected. At one extreme, if the gradient noise was completely random, i.e., if $\rho_i \equiv 1/2$, then the $\rho$-norm would become identical zero for any gradient vector and rates would be trivial inequalities, leading to divergence as in the counterexample. At other extreme, if there was no gradient noise, i.e., if $\rho_i \equiv 1$, then the $\rho$-norm would be just the $l^1$ norm and from (6) we get the rate $\tilde{\mathcal{O}}(1/\sqrt{K})$ with respect to the $l^1$ norm. However, if we know that $\rho_i > 1/2$, then we can ensure that the method will eventually converge.

• **Geometry.** The presence of the $\rho$-norm in these rates suggests that there is no particular geometry (e.g., $l^1$ or $l^2$) associated with signSGD. Instead, the geometry is induced from the success probabilities. For example, in the case of unbiased and unimodal symmetric noise, the geometry is described by the mixture norm $l^{1,2}$.

• **Practicality.** The rate (7) (as well as (30)) supports the common learning schedule practice of using a constant step size for a period of time, and then halving the step-size and continuing this process.

For a reader interested in comparing Theorem 1 with a standard result for SGD, we state the standard result in the Section C. We now state a general convergence rate for Algorithm 1 with Option 2.

**Theorem 2** (see B.5). *Under the SPB assumption, Algorithm 1 (Option 2) with step sizes $\gamma_k = \gamma_0/\sqrt{k+1}$ converges as follows: $\frac{1}{K}\sum_{k=0}^{K-1} \mathbb{E}\|\nabla f(x_k)\|_\rho \le \frac{1}{\sqrt{K}}\left[\frac{f(x_0)-f^*}{\gamma_0} + \gamma_0 d\bar{L}\right]$. In the case of constant step size $\gamma_k = \gamma > 0$, the same rate as (7) is achieved.*

Comparing Theorem 2 with Theorem 1, notice that a small modification in Algorithm 1 can remove the log-dependent factor from (6); we then bound the average of past gradient norms instead of the minimum. On the other hand, in a big data regime, function evaluations in Algorithm 1 (Option 2, line 4) are infeasible. Clearly, *Option 2* is useful only when one can afford function evaluations and has rough estimates about the gradients (i.e., signs of stochastic gradients). This option should be considered within the framework of derivative-free optimization.

## 4.2 CONVERGENCE ANALYSIS IN DISTRIBUTED SETTING

In this part we present the convergence result of distributed signSGD (*Algorithm 2*) with majority vote introduced in Bernstein et al. (2018). Majority vote is considered within a parameter server framework, where for each coordinate parameter server receives one sign from each node and sends

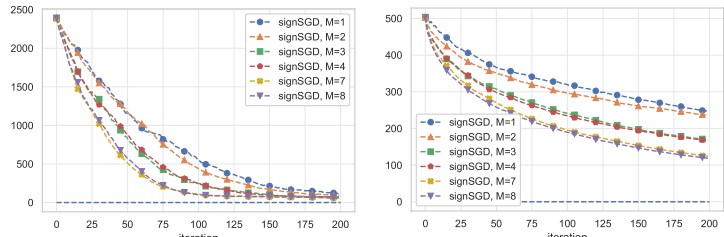

Figure 2: Experiments on distributed signSGD with majority vote using Rosenbrock function. Plots show function values with respect to iterations averaged over 10 repetitions. Left plot used constant step size $\gamma = 0.02$, right plot used variable step size with $\gamma_0 = 0.02$. We set mini-batch size 1 and used the same initial point. Dashed blue lines show the minimum value.

back the sign sent by the majority of nodes. Known convergence results (Bernstein et al., 2018; 2019) use $O(K)$ mini-batch size as well as $O(1/K)$ constant step size. In the sequel we remove this limitations extending Theorem 1 to distributed training. In distributed setting the number of nodes $M$ get involved in geometry introducing new $\rho_M$-norm, which is defined by the regularized incomplete beta function $I$ (see B.6).

---

**Algorithm 2** DISTRIBUTED SIGNSGD WITH MAJORITY VOTE

1: **Input:** step sizes $\{\gamma_k\}$, current point $x_k$, # of nodes $M$
2: **on** each node
3:      $\hat{g}^m(x_k) \leftarrow$ StochasticGradient$(x_k)$
4: **on** server
5:      **pull** sign $\hat{g}^m(x_k)$ **from** each node
6:      **push** sign $\left[ \sum_{m=1}^{M} \text{sign} \, \hat{g}^m(x_k) \right]$ **to** each node
7: **on** each node
8:      $x_{k+1} \leftarrow x_k - \gamma_k \, \text{sign} \left[ \sum_{m=1}^{M} \text{sign} \, \hat{g}^m(x_k) \right]$

---

**Definition 2** ($\rho_M$-norm). *Let $M \geq 1$ be the number of nodes and $l = \left[ \frac{M+1}{2} \right]$. Define $\rho_M$-norm of gradient $g(x)$ at $x \in \mathbb{R}^d$ as $\|g(x)\|_{\rho_M} = \sum_{i=1}^{d} \left( 2I(\rho_i(x); l, l) - 1 \right) |g_i(x)|$.*

Now we can state the convergence rate of distributed signSGD with majority vote.

**Theorem 3** (Non-convex convergence of distributed signSGD, see B.6). *Under SPB assumption, distributed signSGD (Algorithm 2) with step sizes $\gamma_k = \gamma_0 / \sqrt{k+1}$ converges as follows*

$$\min_{0 \leq k < K} \mathbb{E} \|\nabla f(x_k)\|_{\rho_M} \leq \frac{1}{\sqrt{K}} \left[ \frac{f(x_0) - f^*}{\gamma_0} + \gamma_0 d\bar{L} \right] + \frac{\gamma_0 d\bar{L}}{2} \frac{\log K}{\sqrt{K}}. \tag{8}$$

*For constant step sizes $\gamma_k \equiv \gamma > 0$, we have convergence up to a level proportional to step size $\gamma$:*

$$\frac{1}{K} \sum_{k=0}^{K-1} \mathbb{E} \|\nabla f(x_k)\|_{\rho_M} \leq \frac{f(x_0) - f^*}{\gamma K} + \frac{d\bar{L}}{2} \gamma. \tag{9}$$

**Variance Reduction.** Using Hoeffding's inequality, we show that $\|g(x)\|_{\rho_M} \to \|g(x)\|_1$ exponentially fast as $M \to \infty$: $\left( 1 - \exp\left( -(2\rho(x) - 1)^2 l \right) \right) \|g(x)\|_1 \leq \|g(x)\|_{\rho_M} \leq \|g(x)\|_1$, where $\rho(x) = \min_{1 \leq i \leq d} \rho_i(x) > \frac{1}{2}$. Hence, in some sense, we have exponential variance reduction in terms of number of nodes (see B.7).

**Number of nodes.** Notice that theoretically there is no difference between $2l-1$ and $2l$ nodes, and this in not a limitation of the analysis. Indeed, as it is shown in the proof, expected sign vector at the master with $M = 2l - 1$ nodes is the same as with $M = 2l$ nodes: $\mathbb{E} \, \text{sign}(\hat{g}_i^{(2l)} \cdot g_i) = \mathbb{E} \, \text{sign}(\hat{g}_i^{(2l-1)} \cdot g_i)$, where $\hat{g}^{(M)}$ is the sum of stochastic sign vectors aggregated from nodes. The intuition behind this phenomenon is that majority vote with even number of nodes, e.g. $M = 2l$, fails to provide any sign

with little probability (it is the probability of half nodes voting for $+1$, and half nodes voting for $-1$). However, if we remove one node, e.g. $M = 2l - 1$, then master receives one sign-vote less but gets rid of that little probability of failing the vote (sum of odd number of $\pm 1$ cannot vanish). So, somehow this two things cancel each other and we gain no improvement in expectation adding one more node to parameter server framework with odd number of nodes.

## 5  EXPERIMENTS

We verify our theoretical results experimentally using the MNIST dataset with feed-forward neural network (FNN) and the well known Rosenbrock (non-convex) function with $d = 10$ variables:

$$f(x) = \sum_{i=1}^{d-1} f_i(x) = \sum_{i=1}^{d-1} 100(x_{i+1} - x_i^2)^2 + (1 - x_i)^2, \quad x \in \mathbb{R}^d. \tag{10}$$

Stochastic formulation of minimization problem for Rosenbrock function is as follows: at any point $x \in \mathbb{R}^d$ we have access to *biased* stochastic gradient $\hat{g}(x) = \nabla f_i(x) + \xi$, where index $i$ is chosen uniformly at random from $\{1, 2, \ldots, d-1\}$ and $\xi \sim \mathcal{N}(0, \nu^2 I)$ with $\nu > 0$.

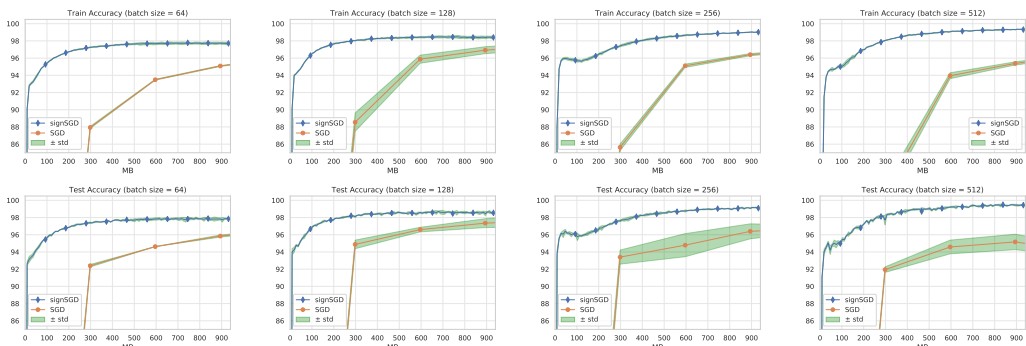

Figure 3: Comparison of signSGD and SGD on MNIST dataset with a fixed budget of gradient communication (MB) using single hidden layer FNN. For each batch size, we first tune the constant step size over logarithmic scale $\{10, 1, 0.1, 0.01, 0.001\}$ and then fine tune it. Clearly, signSGD beats SGD if we compare their accuracies against communication. As suggested by the theory (see Lemma 3) bigger mini-batch size increases the success probabilities $\rho_i$ and thus improves the convergence.

Figure 2 illustrates the effect of multiple nodes in distributed training with majority vote. As we see increasing the number of nodes improves the convergence rate. It also supports the claim that in expectation there is no improvement from $2l - 1$ nodes to $2l$ nodes.

Figure 4 shows the robustness of SPB assumption in the convergence rate (7) with constant step size. We exploited four levels of noise in each column to demonstrate the correlation between success probabilities and convergence rate. In the first experiment (first column) SPB assumption is violated strongly and the corresponding rate shows divergence. In the second column, probabilities still violating SPB assumption are close to the threshold and the rate shows oscillations. Next columns show the improvement in rates when success probabilities are pushed to be close to 1.

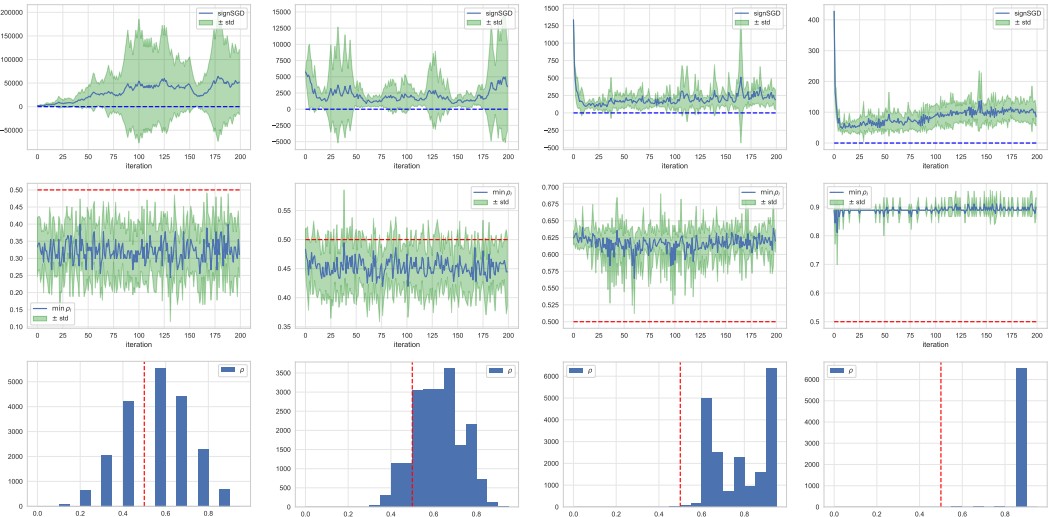

Figure 4: Performance of signSGD with constant step size ($\gamma = 0.25$) under four different noise levels (mini-batch size 1, 2, 5, 8) using Rosenbrock function. Each column represent a separate experiment with function values, evolution of minimum success probabilities and the histogram of success probabilities throughout the iteration process. Dashed blue line in the first row is the minimum value. Dashed red lines in second and third rows are thresholds $1/2$ of success probabilities. The shaded area in first and second rows shows standard deviation obtained from ten repetitions.

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

# Appendix: "On Stochastic Sign Descent Methods"

## A    EXTRA EXPERIMENTS

In this section we perform several additional experiments for further insights.

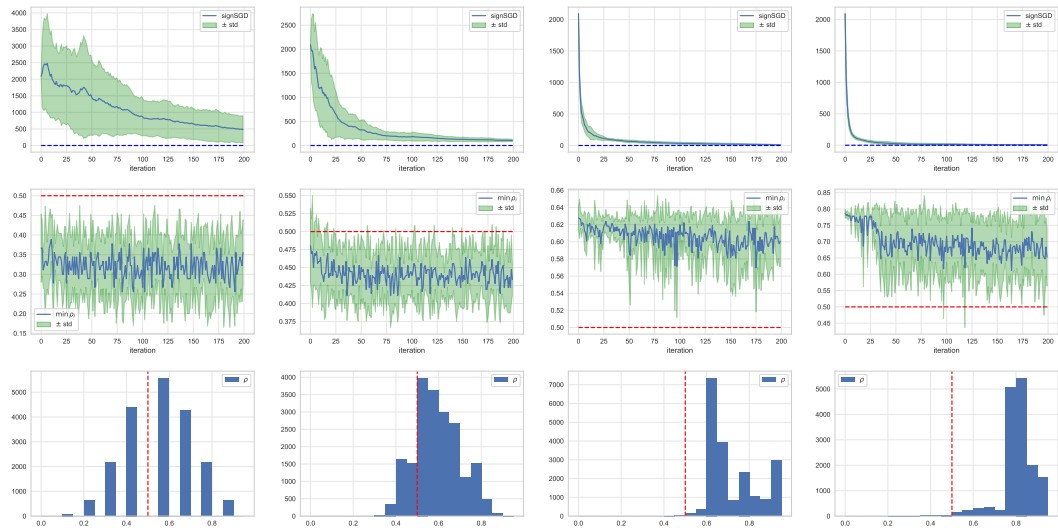

Figure 5: Performance of signSGD with variable step size ($\gamma_0 = 0.25$) under four different noise levels (mini-batch size 1, 2, 5, 7) using Rosenbrock function. As in the experiments of Figure 4 with constant step size, these plots show the relationship between success probabilities and the convergence rate (6). In low success probability regime (first and second columns) we observe oscillations, while in high success probability regime (third and forth columns) oscillations are mitigated substantially.

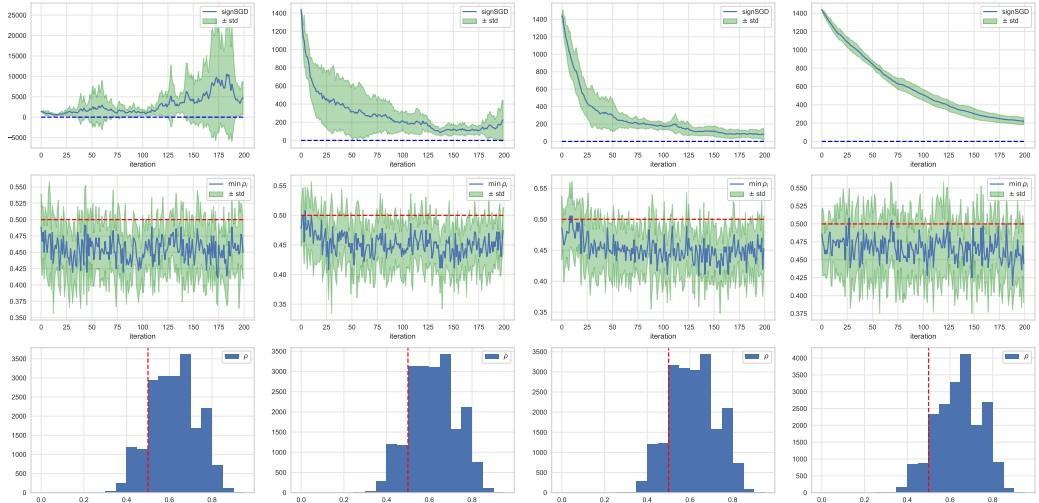

Figure 6: In this part of experiments we investigated convergence rate (7) to a neighborhood of the solution. We fixed gradient noise level by setting mini-batch size 2 and altered the constant step size. For the first column we set bigger step size $\gamma = 0.25$ to detect the divergence (as we slightly violated SPB assumption). Then for the second and third columns we set $\gamma = 0.1$ and $\gamma = 0.05$ to expose the convergence to a neighborhood of the minimizer. For the forth column we set even smaller step size $\gamma = 0.01$ to observe a slower convergence.

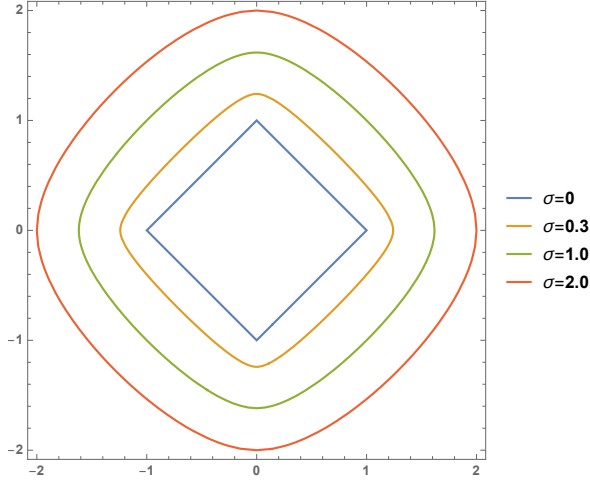

Figure 7: Unit balls in $l^{1,2}$ norm (5) with different noise levels.

# B    PROOFS

## B.1    SUFFICIENT CONDITIONS FOR SPB: PROOF OF LEMMA 1

Here we state the well-known Gauss's inequality on unimodal distributions[3].

**Theorem 4** (Gauss's inequality). *Let $X$ be a unimodal random variable with mode $m$, and let $\sigma_m^2$ be the expected value of $(X - m)^2$. Then for any positive value of $r$,*

$$\mathrm{Prob}(|X - m| > r) \leq \begin{cases} \frac{4}{9}\left(\frac{\sigma_m}{r}\right)^2, & \text{if } r \geq \frac{2}{\sqrt{3}}\sigma_m \\ 1 - \frac{1}{\sqrt{3}}\frac{r}{\sigma_m}, & \text{otherwise} \end{cases}$$

Applying this inequality on unimodal and symmetric distributions, direct algebraic manipulations give the following bound:

$$\mathrm{Prob}(|X - \mu| \leq r) \geq \begin{cases} 1 - \frac{4}{9}\left(\frac{\sigma}{r}\right)^2, & \text{if } \frac{\sigma}{r} \leq \frac{\sqrt{3}}{2} \\ \frac{1}{\sqrt{3}}\frac{r}{\sigma}, & \text{otherwise} \end{cases} \geq \frac{r/\sigma}{r/\sigma + \sqrt{3}},$$

where $m = \mu$ and $\sigma_m^2 = \sigma^2$ are the mean and variance of unimodal, symmetric random variable $X$, and $r \geq 0$. Now, using the assumption that each $\hat{g}_i(x)$ has unimodal and symmetric distribution, we apply this bound for $X = \hat{g}_i(x)$, $\mu = g_i(x)$, $\sigma^2 = \sigma_i^2(x)$ and get a bound for success probabilities

$$\mathrm{Prob}(\mathrm{sign}\,\hat{g}_i = \mathrm{sign}\,g_i) = \begin{cases} \mathrm{Prob}(\hat{g}_i \geq 0), & \text{if } g_i > 0 \\ \mathrm{Prob}(\hat{g}_i \leq 0), & \text{if } g_i < 0 \end{cases}$$

$$= \begin{cases} \frac{1}{2} + \mathrm{Prob}(0 \leq \hat{g}_i \leq g_i), & \text{if } g_i > 0 \\ \frac{1}{2} + \mathrm{Prob}(g_i \leq \hat{g}_i \leq 0), & \text{if } g_i < 0 \end{cases}$$

$$= \begin{cases} \frac{1}{2} + \frac{1}{2}\mathrm{Prob}(0 \leq \hat{g}_i \leq 2g_i), & \text{if } g_i > 0 \\ \frac{1}{2} + \frac{1}{2}\mathrm{Prob}(2g_i \leq \hat{g}_i \leq 0), & \text{if } g_i < 0 \end{cases}$$

$$= \frac{1}{2} + \frac{1}{2}\mathrm{Prob}(|\hat{g}_i - g_i| \leq |g_i|)$$

$$\geq \frac{1}{2} + \frac{1}{2}\frac{|g_i|/\sigma_i}{|g_i|/\sigma_i + \sqrt{3}}$$

$$= \frac{1}{2} + \frac{1}{2}\frac{|g_i|}{|g_i| + \sqrt{3}\sigma_i}$$

**Improvment on Lemma 1 and $l^{1,2}$ norm:** The bound after Gauss inequality can be improved including a second order term

$$\mathrm{Prob}(|X - \mu| \leq r) \geq \begin{cases} 1 - \frac{4}{9}\left(\frac{\sigma}{r}\right)^2, & \text{if } \frac{\sigma}{r} \leq \frac{\sqrt{3}}{2} \\ \frac{1}{\sqrt{3}}\frac{r}{\sigma}, & \text{otherwise} \end{cases} \geq 1 - \frac{1}{1 + r/\sqrt{3}\sigma + (r/\sqrt{3}\sigma)^2}.$$

Indeed, letting $z := {r}/{\sqrt{3}\sigma} \geq {2}/{3}$, we get $1 - \frac{4}{9}\frac{1}{3z^2} \geq 1 - \frac{1}{1+z+z^2}$ as it reduces to $23z^2 - 4z - 4 \geq 0$. Otherwise, if $0 \leq z \leq {2}/{3}$, then $z \geq 1 - \frac{1}{1+z+z^2}$ as it reduces to $1 \geq 1 - z^3$. The improvement is tighter as

$$\frac{r/\sigma}{r/\sigma + \sqrt{3}} = 1 - \frac{1}{1 + r/\sqrt{3}\sigma} \leq 1 - \frac{1}{1 + r/\sqrt{3}\sigma + (r/\sqrt{3}\sigma)^2}.$$

Hence, continuing the proof of Lemma 1, we get

$$\mathrm{Prob}(\mathrm{sign}\,\hat{g}_i = \mathrm{sign}\,g_i) \geq 1 - \frac{1}{2}\frac{1}{1 + |g_i|/\sqrt{3}\sigma_i + (|g_i|/\sqrt{3}\sigma_i)^2}$$

and we could have defined $l^{1,2}$-norm in a bit more complicated form as

$$\|g\|_{l^{1,2}} := \sum_{i=1}^{d}\left(1 - \frac{1}{1 + |g_i|/\sqrt{3}\sigma_i + (|g_i|/\sqrt{3}\sigma_i)^2}\right)|g_i|.$$

---

[3]see https://en.wikipedia.org/wiki/Gauss%27s_inequality

## B.2    SUFFICIENT CONDITIONS FOR SPB: PROOF OF LEMMA 2

Let $\hat{g}^{(\tau)}$ be the gradient estimator with mini-batch size $\tau$. It is known that the variance for $\hat{g}^{(\tau)}$ is dropped by at least a factor of $\tau$, i.e.

$$\mathbb{E}[(\hat{g}_i^{(\tau)} - g_i)^2] \leq \frac{\sigma_i^2}{\tau}.$$

Hence, estimating the failure probabilities of sign $\hat{g}^{(\tau)}$ when $g_i \neq 0$, we have

$$\begin{aligned}
\text{Prob}(\text{sign}\,\hat{g}_i^{(\tau)} \neq \text{sign}\,g_i) &= \text{Prob}(|\hat{g}_i^{(\tau)} - g_i| = |\hat{g}_i^{(\tau)}| + |g_i|) \\
&\leq \text{Prob}(|\hat{g}_i^{(\tau)} - g_i| \geq |g_i|) \\
&= \text{Prob}((\hat{g}_i^{(\tau)} - g_i)^2 \geq g_i^2) \\
&\leq \frac{\mathbb{E}[(\hat{g}_i^{(\tau)} - g_i)^2]}{g_i^2} \\
&= \frac{\sigma_i^2}{\tau g_i^2},
\end{aligned}$$

which imples

$$\rho_i = \text{Prob}(\text{sign}\,\hat{g}_i = \text{sign}\,g_i) \geq 1 - \frac{\sigma_i^2}{\tau g_i^2} \geq 1 - \frac{c}{\tau}.$$

## B.3    SUFFICIENT CONDITIONS FOR SPB: PROOF OF LEMMA 3

We will split the derivation into three lemmas providing some intuition on the way. The first two lemmas establish success probability bounds in terms of mini-batch size. Essentially, we present two methods: one works well in the case of small randomness, while the other one in the case of non-small randomness. In the third lemma, we combine those two bounds to get the condition on mini-batch size ensuring SPB assumption.

**Lemma 4.** *Let $X_1, X_2, \ldots, X_\tau$ be i.i.d. random variables with non-zero mean $\mu := \mathbb{E}X_1 \neq 0$, finite variance $\sigma^2 := \mathbb{E}|X_1 - \mu|^2 < \infty$. Then for any mini-batch size $\tau \geq 1$*

$$\text{Prob}\left(\text{sign}\left[\frac{1}{\tau}\sum_{i=1}^{\tau}X_i\right] = \text{sign}\,\mu\right) \geq 1 - \frac{\sigma^2}{\tau\mu^2}. \tag{11}$$

*Proof.* Without loss of generality, we assume $\mu > 0$. Then, after some adjustments, the proof follows from the Chebyshev's inequality:

$$\begin{aligned}
\text{Prob}\left(\text{sign}\left[\frac{1}{\tau}\sum_{i=1}^{\tau}X_i\right] = \text{sign}\,\mu\right) &= \text{Prob}\left(\frac{1}{\tau}\sum_{i=1}^{\tau}X_i > 0\right) \\
&\geq \text{Prob}\left(\left|\frac{1}{\tau}\sum_{i=1}^{\tau}X_i - \mu\right| < \mu\right) \\
&= 1 - \text{Prob}\left(\left|\frac{1}{\tau}\sum_{i=1}^{\tau}X_i - \mu\right| \geq \mu\right) \\
&\geq 1 - \frac{1}{\mu^2}\,\text{Var}\left[\frac{1}{\tau}\sum_{i=1}^{\tau}X_i\right] \\
&= 1 - \frac{\sigma^2}{\tau\mu^2},
\end{aligned}$$

where in the last step we used independence of random variables $X_1, X_2, \ldots, X_\tau$. $\qquad\square$

Obviously, bound (11) is not optimal for big variance as it becomes a trivial inequality. In the case of non-small randomness a better bound is achievable additionally assuming the finitness of 3th central moment.

**Lemma 5.** *Let $X_1, X_2, \ldots, X_\tau$ be i.i.d. random variables with non-zero mean $\mu := \mathbb{E}X_1 \neq 0$, positive variance $\sigma^2 := \mathbb{E}|X_1 - \mu|^2 > 0$ and finite 3th central moment $\nu^3 := \mathbb{E}|X_1 - \mu|^3 < \infty$. Then for any mini-batch size $\tau \geq 1$*

$$\text{Prob}\left(\text{sign}\left[\frac{1}{\tau}\sum_{i=1}^{\tau}X_i\right] = \text{sign}\,\mu\right) \geq \frac{1}{2}\left(1 + \text{erf}\left(\frac{|\mu|\sqrt{\tau}}{\sqrt{2}\sigma}\right) - \frac{\nu^3}{\sigma^3\sqrt{\tau}}\right), \tag{12}$$

*where error function* erf *is defined as*

$$\text{erf}(x) = \frac{2}{\sqrt{\pi}}\int_0^x e^{-t^2}\,dt, \quad x \in \mathbb{R}.$$

*Proof.* Again, without loss of generality, we may assume that $\mu > 0$. Informally, the proof goes as follows. As we have an average of i.i.d. random variables, we approximate it (in the sense of distribution) by normal distribution using the Central Limit Theorem (CLT). Then we compute success probabilities for normal distribution with the error function erf. Finally, we take into account the approximation error in CLT, from which the third term with negative sign appears. More formally, we apply Berry–Esseen inequality[4] on the rate of approximation in CLT (Shevtsova, 2011):

$$\left|\text{Prob}\left(\frac{1}{\sigma\sqrt{\tau}}\sum_{i=1}^{\tau}(X_i - \mu) > t\right) - \text{Prob}\left(N > t\right)\right| \leq \frac{1}{2}\frac{\nu^3}{\sigma^3\sqrt{\tau}}, \quad t \in \mathbb{R},$$

where $N \sim \mathcal{N}(0,1)$ has the standard normal distribution. Setting $t = -\mu\sqrt{\tau}/\sigma$, we get

$$\left|\text{Prob}\left(\frac{1}{\tau}\sum_{i=1}^{\tau}X_i > 0\right) - \text{Prob}\left(N > -\frac{\mu\sqrt{\tau}}{\sigma}\right)\right| \leq \frac{1}{2}\frac{\nu^3}{\sigma^3\sqrt{\tau}}. \tag{13}$$

It remains to compute the second probability using the cumulative distribution function of normal distribuition and express it in terms of the error function:

$$\begin{aligned}
\text{Prob}\left(\text{sign}\left[\frac{1}{\tau}\sum_{i=1}^{\tau}X_i\right] = \text{sign}\,\mu\right) &= \text{Prob}\left(\frac{1}{\tau}\sum_{i=1}^{\tau}X_i > 0\right) \\
&\overset{(13)}{\geq} \text{Prob}\left(N > -\frac{\mu\sqrt{\tau}}{\sigma}\right) - \frac{1}{2}\frac{\nu^3}{\sigma^3\sqrt{\tau}} \\
&= \frac{1}{\sqrt{2\pi}}\int_{-\mu\sqrt{\tau}/\sigma}^{\infty} e^{-t^2/2}\,dt - \frac{1}{2}\frac{\nu^3}{\sigma^3\sqrt{\tau}} \\
&= \frac{1}{2}\left(1 + \sqrt{\frac{2}{\pi}}\int_0^{\mu\sqrt{\tau}/\sigma} e^{-t^2/2}\,dt - \frac{\nu^3}{\sigma^3\sqrt{\tau}}\right) \\
&= \frac{1}{2}\left(1 + \text{erf}\left(\frac{\mu\sqrt{\tau}}{\sqrt{2}\sigma}\right) - \frac{\nu^3}{\sigma^3\sqrt{\tau}}\right).
\end{aligned}$$

$\square$

Clearly, bound (12) is better than (11) when randomness is high. On the other hand, bound (12) is not optimal for small randomness ($\sigma \approx 0$). Indeed, one can show that in a small randomness regime, while both variance $\sigma^2$ and third moment $\nu^3$ are small, the ration $\nu/\sigma$ might blow up to infinity producing trivial inequality. For instance, taking $X_i \sim \text{Bernoulli}(p)$ and letting $p \to 1$ gives $\nu/\sigma = O\left((1-p)^{-1/6}\right)$. This behaviour stems from the fact that we are using CLT: less randomness implies slower rate of approximation in CLT.

As a result of these two bounds on success probabilities, we conclude a condition on mini-batch size for the SPB assumption to hold.

---

[4]see https://en.wikipedia.org/wiki/Berry-Esseen_theorem

**Lemma 6.** *Let $X_1, X_2, \ldots, X_\tau$ be i.i.d. random variables with non-zero mean $\mu \neq 0$ and finite variance $\sigma^2 < \infty$. Then*

$$\text{Prob}\left(\text{sign}\left[\frac{1}{\tau}\sum_{i=1}^{\tau}X_i\right] = \text{sign}\,\mu\right) > \frac{1}{2}, \quad \text{if} \quad \tau > 2\min\left(\frac{\sigma^2}{\mu^2}, \frac{\nu^3}{|\mu|\sigma^2}\right), \tag{14}$$

*where $\nu^3$ is (possibly infinite) 3th central moment.*

*Proof.* First, if $\sigma = 0$ then the lemma holds trivially. If $\nu = \infty$, then it follows immediately from Lemma 4. Assume both $\sigma$ and $\nu$ are positive and finite.

In case of $\tau > 2\sigma^2/\mu^2$ we apply Lemma 4 again. Consider the case $\tau \leq 2\sigma^2/\mu^2$, which implies $\frac{\mu\sqrt{\tau}}{\sqrt{2}\sigma} \leq 1$. It is easy to check that $\text{erf}(x)$ is concave on $[0, 1]$ (in fact on $[0, \infty)$), therefore $\text{erf}(x) \geq \text{erf}(1)x$ for any $x \in [0, 1]$. Setting $x = \frac{\mu\sqrt{\tau}}{\sqrt{2}\sigma}$ we get

$$\text{erf}\left(\frac{\mu\sqrt{\tau}}{\sqrt{2}\sigma}\right) \geq \frac{\text{erf}(1)}{\sqrt{2}}\frac{\mu\sqrt{\tau}}{\sigma},$$

which together with (12) gives

$$\text{Prob}\left(\text{sign}\left[\frac{1}{\tau}\sum_{i=1}^{\tau}X_i\right] = \text{sign}\,\mu\right) \geq \frac{1}{2}\left(1 + \frac{\text{erf}(1)}{\sqrt{2}}\frac{\mu\sqrt{\tau}}{\sigma} - \frac{\nu^3}{\sigma^3\sqrt{\tau}}\right).$$

Hence, SPB assumption holds if

$$\tau > \frac{\sqrt{2}}{\text{erf}(1)}\frac{\nu^3}{\mu\sigma^2}.$$

It remains to show that $\text{erf}(1) > 1/\sqrt{2}$. Convexity of $e^x$ on $x \in [-1, 0]$ implies $e^x \geq 1 + (1 - 1/e)x$ for any $x \in [-1, 0]$. Therefore

$$\begin{aligned}
\text{erf}(1) &= \frac{2}{\sqrt{\pi}}\int_0^1 e^{-t^2}\,dt \\
&\geq \frac{2}{\sqrt{\pi}}\int_0^1\left(1 - (1 - 1/e)t^2\right)\,dt \\
&= \frac{2}{\sqrt{\pi}}\left(\frac{2}{3} + \frac{1}{3e}\right) > \frac{2}{\sqrt{4}}\left(\frac{2}{3} + \frac{1}{3 \cdot 3}\right) = \frac{7}{9} > \frac{1}{\sqrt{2}}.
\end{aligned}$$

$\square$

Lemma (3) follows from Lemma (6) applying it to i.i.d. data $\hat{g}_i^1(x), \hat{g}_i^2(x), \ldots, \hat{g}_i^M(x)$.

### B.4 CONVERGENCE ANALYSIS: PROOF OF THEOREM 1

First, from $L$-smoothness assumption we have

$$\begin{aligned}
f(x_{k+1}) &= f(x_k - \gamma_k\,\text{sign}\,\hat{g}_k) \\
&\leq f(x_k) - \langle g_k, \gamma_k\,\text{sign}\,\hat{g}_k\rangle + \sum_{i=1}^{d}\frac{L_i}{2}(\gamma_k\,\text{sign}\,\hat{g}_{k,i})^2 \\
&= f(x_k) - \gamma_k\langle g_k, \text{sign}\,\hat{g}_k\rangle + \frac{d\bar{L}}{2}\gamma_k^2,
\end{aligned}$$

where $g_k = g(x_k)$, $\hat{g}_k = \hat{g}(x_k)$, $\hat{g}_{k,i}$ is the $i$-th component of $\hat{g}_k$ and $\bar{L}$ is the average value of $L_i$'s. Taking conditional expectation given current iteration $x_k$ gives

$$\mathbb{E}[f(x_{k+1})|x_k] \leq f(x_k) - \gamma_k\mathbb{E}[\langle g_k, \text{sign}\,\hat{g}_k\rangle] + \frac{d\bar{L}}{2}\gamma_k^2. \tag{15}$$

Using the definition of success probabilities $\rho_i$ we get

$$\mathbb{E}[\langle g_k, \operatorname{sign} \hat{g}_k \rangle] = \langle g_k, \mathbb{E}[\operatorname{sign} \hat{g}_k] \rangle \tag{16}$$

$$= \sum_{i=1}^{d} g_{k,i} \cdot \mathbb{E}[\operatorname{sign} \hat{g}_{k,i}] = \sum_{\substack{1 \le i \le d \\ g_{k,i} \ne 0}} g_{k,i} \cdot \mathbb{E}[\operatorname{sign} \hat{g}_{k,i}] \tag{17}$$

$$= \sum_{\substack{1 \le i \le d \\ g_{k,i} \ne 0}} g_{k,i} \left( \rho_i(x_k) \operatorname{sign} g_{k,i} + (1 - \rho_i(x_k))(-\operatorname{sign} g_{k,i}) \right) \tag{18}$$

$$= \sum_{\substack{1 \le i \le d \\ g_{k,i} \ne 0}} (2\rho_i(x_k) - 1)|g_{k,i}| = \sum_{i=1}^{d} (2\rho_i(x_k) - 1)|g_{k,i}| = \|g_k\|_\rho. \tag{19}$$

Plugging this into (15) and taking full expectation, we get

$$\mathbb{E}\|g_k\|_\rho \le \frac{\mathbb{E}[f(x_k)] - \mathbb{E}[f(x_{k+1})]}{\gamma_k} + \frac{d\bar{L}}{2} \gamma_k. \tag{20}$$

Therefore

$$\sum_{k=0}^{K-1} \gamma_k \mathbb{E}\|g_k\|_\rho \le (f(x_0) - f^*) + \frac{d\bar{L}}{2} \sum_{k=0}^{K-1} \gamma_k^2. \tag{21}$$

Now, in case of decreasing step sizes $\gamma_k = \gamma_0/\sqrt{k+1}$

$$\begin{aligned}
\min_{0 \le k < K} \mathbb{E}\|g_k\|_\rho &\le \sum_{k=0}^{K-1} \frac{\gamma_0}{\sqrt{k+1}} \mathbb{E}\|g_k\|_\rho \bigg/ \sum_{k=0}^{K-1} \frac{\gamma_0}{\sqrt{k+1}} \\
&\le \frac{1}{\sqrt{K}} \left[ \frac{f(x_0) - f^*}{\gamma_0} + \frac{d\bar{L}}{2} \gamma_0 \sum_{k=0}^{K-1} \frac{1}{k+1} \right] \\
&\le \frac{1}{\sqrt{K}} \left[ \frac{f(x_0) - f^*}{\gamma_0} + \gamma_0 d\bar{L} + \frac{\gamma_0 d\bar{L}}{2} \log K \right] \\
&= \frac{1}{\sqrt{K}} \left[ \frac{f(x_0) - f^*}{\gamma_0} + \gamma_0 d\bar{L} \right] + \frac{\gamma_0 d\bar{L}}{2} \frac{\log K}{\sqrt{K}}.
\end{aligned}$$

where we have used the following standard inequalities

$$\sum_{k=1}^{K} \frac{1}{\sqrt{k}} \ge \sqrt{K}, \quad \sum_{k=1}^{K} \frac{1}{k} \le 2 + \log K. \tag{22}$$

In the case of constant step size $\gamma_k = \gamma$

$$\frac{1}{K} \sum_{k=0}^{K-1} \mathbb{E}\|g_k\|_\rho \le \frac{1}{\gamma K} \left[ (f(x_0) - f^*) + \frac{d\bar{L}}{2} \gamma^2 K \right] = \frac{f(x_0) - f^*}{\gamma K} + \frac{d\bar{L}}{2} \gamma.$$

### B.5 CONVERGENCE ANALYSIS: PROOF OF THEOREM 2

Clearly, the iterations $\{x_k\}_{k\geq 0}$ of Algorithm 1 (Option 2) do not increase the function value in any iteration, i.e. $\mathbb{E}[f(x_{k+1})|x_k] \leq f(x_k)$. Continuing the proof of Theorem 1 from (20), we get

$$
\begin{aligned}
\frac{1}{K}\sum_{k=0}^{K-1}\mathbb{E}\|g_k\|_\rho &\leq \frac{1}{K}\sum_{k=0}^{K-1}\frac{\mathbb{E}[f(x_k)]-\mathbb{E}[f(x_{k+1})]}{\gamma_k}+\frac{d\bar{L}}{2}\gamma_k\\
&=\frac{1}{K}\sum_{k=0}^{K-1}\frac{\mathbb{E}[f(x_k)]-\mathbb{E}[f(x_{k+1})]}{\gamma_0}\sqrt{k+1}+\frac{d\bar{L}}{2K}\sum_{k=0}^{K-1}\frac{\gamma_0}{\sqrt{k+1}}\\
&\leq\frac{1}{\sqrt{K}}\sum_{k=0}^{K-1}\frac{\mathbb{E}[f(x_k)]-\mathbb{E}[f(x_{k+1})]}{\gamma_0}+\frac{\gamma_0 d\bar{L}}{\sqrt{K}}\\
&=\frac{f(x_0)-\mathbb{E}[f(x_K)]}{\gamma_0\sqrt{K}}+\frac{\gamma_0 d\bar{L}}{\sqrt{K}}\\
&\leq\frac{1}{\sqrt{K}}\left[\frac{f(x_0)-f^*}{\gamma_0}+\gamma_0 d\bar{L}\right],
\end{aligned}
$$

where we have used the following inequality

$$
\sum_{k=1}^{K}\frac{1}{\sqrt{k}}\leq 2\sqrt{K}.
$$

The proof for constant step size is the same as in Theorem 1.

### B.6 CONVERGENCE ANALYSIS IN DISTRIBUTED SETTING: PROOF OF THEOREM 3

First, denote by $I(p;a,b)$ the regularized incomplete beta function, which is defined as follows

$$
I(p;a,b)=\frac{B(p;a,b)}{B(a,b)}=\frac{\int_0^p t^{a-1}(1-t)^{b-1}\,dt}{\int_0^1 t^{a-1}(1-t)^{b-1}\,dt},\quad a,b>0,\ p\in[0,1]. \tag{23}
$$

The proof of Theorem 3 goes with the same steps as in Theorem 1, except the derivation (16)–(19) is replaced by

$$
\begin{aligned}
\mathbb{E}[\langle g_k,\operatorname{sign}\hat{g}_k^{(M)}\rangle] &=\langle g_k,\mathbb{E}[\operatorname{sign}\hat{g}_k^{(M)}]\rangle\\
&=\sum_{i=1}^{d}g_{k,i}\cdot\mathbb{E}[\operatorname{sign}\hat{g}_{k,i}^{(M)}]\\
&=\sum_{\substack{1\leq i\leq d\\ g_{k,i}\neq 0}}|g_{k,i}|\cdot\mathbb{E}\left[\operatorname{sign}\left(\hat{g}_{k,i}^{(M)}\cdot g_{k,i}\right)\right]\\
&=\sum_{\substack{1\leq i\leq d\\ g_{k,i}\neq 0}}|g_{k,i}|\left(2I(\rho_i(x_k);l,l)-1\right)=\|g_k\|_{\rho_M},
\end{aligned}
$$

where we have used the following lemma.

**Lemma 7.** *Assume that for some point $x\in\mathbb{R}^d$ and some coordinate $i\in\{1,2,\dots,d\}$, master node receives $M$ independent stochastic signs $\operatorname{sign}\hat{g}_i^m(x)$, $m=1,\dots,M$ of true gradient $g_i(x)\neq 0$. Let $\hat{g}^{(M)}(x)$ be the sum of stochastic signs aggregated from nodes:*

$$
\hat{g}^{(M)}=\sum_{m=1}^{M}\operatorname{sign}\hat{g}^m.
$$

*Then*

$$
\mathbb{E}\left[\operatorname{sign}\left(\hat{g}_i^{(M)}\cdot g_i\right)\right]=2I(\rho_i;l,l)-1, \tag{24}
$$

*where $l=\lceil (M+1)/2\rceil$ and $\rho_i>1/2$ is the success probablity for coordinate $i$.*

*Proof.* Denote by $S_i^m$ the Bernoulli trial of node $m$ corresponding to $i$th coordinate, where "success" is the sign match between stochastic gradient and gradient:

$$S_i^m := \begin{cases} 1, & \text{if } \operatorname{sign} \hat{g}_i^m = \operatorname{sign} g_i \\ 0, & \text{otherwise} \end{cases} \sim \text{Bernoulli}(\rho_i). \tag{25}$$

Since nodes have their own independent stochastic gradients and the objective function (or dataset) is shared, then master node receives i.i.d. trials $S_i^m$, which sum up to a binomial random variable $S_i$:

$$S_i := \sum_{m=1}^{M} S_i^m \sim \text{Binomial}(M, \rho_i). \tag{26}$$

First, let us consider the case when there are odd number of nodes, i.e. $M = 2l - 1$, $l \geq 1$. In this case, taking into account (25) and (26), we have

$$\text{Prob}\left(\operatorname{sign} \hat{g}_i^{(M)} = 0\right) = 0,$$

$$\rho_i^{(M)} := \text{Prob}\left(\operatorname{sign} \hat{g}_i^{(M)} = \operatorname{sign} g_i\right) = \text{Prob}(S_i \geq l),$$

$$1 - \rho_i^{(M)} = \text{Prob}\left(\operatorname{sign} \hat{g}_i^{(M)} = -\operatorname{sign} g_i\right).$$

It is well known that cumulative distribution function of binomial random variable can be expressed with regularized incomplete beta function:

$$\text{Prob}(S_i \geq l) = I(\rho_i; l, M - l + 1) = I(\rho_i; l, l). \tag{27}$$

Therefore,

$$\begin{aligned}
\mathbb{E}\left[\operatorname{sign}\left(\hat{g}_i^{(M)} \cdot g_i\right)\right] &= \rho_i^{(M)} \cdot 1 + (1 - \rho_i^{(M)}) \cdot (-1) \\
&= 2\rho_i^{(M)} - 1 \\
&= 2\text{Prob}(S_i \geq l) - 1 \\
&= 2I(\rho_i; l, l) - 1.
\end{aligned}$$

In the case of even number of nodes, i.e. $M = 2l$, $l \geq 1$, there is a probability to fail the vote $\text{Prob}\left(\operatorname{sign} \hat{g}_i^{(M)} = 0\right) > 0$. However using (27) and properties of beta function[5] gives

$$\begin{aligned}
\mathbb{E}\left[\operatorname{sign}\left(\hat{g}_i^{(2l)} \cdot g_i\right)\right] &= \text{Prob}(S_i \geq l + 1) \cdot 1 + \text{Prob}(S_i \leq l - 1) \cdot (-1) \\
&= I(\rho_i; l + 1, l) + I(\rho_i; l, l + 1) - 1 \\
&= 2I(\rho_i; l, l) - 1 \\
&= \mathbb{E}\left[\operatorname{sign}\left(\hat{g}_i^{(2l-1)} \cdot g_i\right)\right].
\end{aligned}$$

This also shows that in expectation there is no difference between having $2l - 1$ and $2l$ nodes. $\qquad\square$

### B.7  CONVERGENCE ANALYSIS IN DISTRIBUTED SETTING: VARIANCE REDUCTION

Here we show exponential variance reduction in distributed setting in terms of number of nodes. We first state the well-known Hoeffding's inequality:

**Theorem 5** (Hoeffding's inequality for general bounded random variables; see (Vershynin, 2018), Theorem 2.2.6). *Let $X_1, X_2, \ldots, X_M$ be independent random variables. Assume that $X_m \in [A_m, B_m]$ for every $m$. Then, for any $t > 0$, we have*

$$\text{Prob}\left(\sum_{m=1}^{M} (X_m - \mathbb{E}X_m) \geq t\right) \leq \exp\left(-\frac{2t^2}{\sum_{m=1}^{M} (B_m - A_m)^2}\right).$$

---

[5]see https://en.wikipedia.org/wiki/Beta_function#Incomplete_beta_function

Define random variables $X_i^m$, $m = 1, 2, \ldots, M$ showing the missmatch between stochastic gradient sign and full gradient sign from node $m$ and coordinate $i$:

$$X_i^m := \begin{cases} -1, & \text{if } \operatorname{sign} \hat{g}_i^m = \operatorname{sign} g_i \\ 1, & \text{otherwise} \end{cases} \tag{28}$$

Clearly $\mathbb{E} X_i^m = 1 - 2\rho_i$ and Hoeffding's inequality gives

$$\operatorname{Prob}\left( \sum_{m=1}^M X_i^m - M(1 - 2\rho_i) \geq t \right) \leq \exp\left( -\frac{t^2}{2M} \right), \quad t > 0.$$

Choosing $t = M(2\rho_i - 1) > 0$ (because of SPB assumption) yields

$$\operatorname{Prob}\left( \sum_{m=1}^M X_i^m \geq 0 \right) \leq \exp\left( -\frac{1}{2}(2\rho_i - 1)^2 M \right).$$

Using Lemma 24, we get

$$2I(\rho_i, l; l) - 1 = \mathbb{E}\left[ \operatorname{sign}\left( \hat{g}_i^{(M)} \cdot g_i \right) \right] = 1 - \operatorname{Prob}\left( \sum_{m=1}^M X_i^m \geq 0 \right) \geq 1 - \exp\left( -(2\rho_i - 1)^2 l \right),$$

which provides the following estimate for $\rho_M$-norm:

$$\left( 1 - \exp\left( -(2\rho(x) - 1)^2 l \right) \right) \|g(x)\|_1 \leq \|g(x)\|_{\rho_M} \leq \|g(x)\|_1,$$

where $\rho(x) = \min_{1 \leq i \leq d} \rho_i(x) > 1/2$.

## C  CONVERGENCE RESULT FOR STANDARD SGD

For comparison, here we state and prove non-convex convergence rates of standard SGD with the same step sizes.

**Theorem 6** (Non-convex convergence of SGD). *Let $\hat{g}$ be an unbiased estimator of the gradient $\nabla f$ and assume that $\mathbb{E}\|\hat{g}\|_2^2 \leq C$ for some $C > 0$. Then SGD with step sizes $\gamma_k = \gamma_0/\sqrt{k+1}$ converges as follows*

$$\min_{0 \leq k < K} \mathbb{E}\|\nabla f(x_k)\|_2^2 \leq \frac{1}{\sqrt{K}}\left[ \frac{f(x_0) - f^*}{\gamma_0} + \gamma_0 C L_{\max} \right] + \frac{\gamma_0 C L_{\max}}{2} \frac{\log K}{\sqrt{K}}. \tag{29}$$

*In the case of constant step size $\gamma_k \equiv \gamma > 0$*

$$\frac{1}{K} \sum_{k=0}^{K-1} \mathbb{E}\|\nabla f(x_k)\|_2^2 \leq \frac{f(x_0) - f^*}{\gamma K} + \frac{C L_{\max}}{2}\gamma. \tag{30}$$

*Proof.* From $L$-smoothness assumption we have

$$\mathbb{E}[f(x_{k+1})|x_k] = \mathbb{E}[f(x_k - \gamma_k \hat{g}_k)|x_k]$$

$$\leq f(x_k) - \mathbb{E}[\langle g_k, \gamma_k \hat{g}_k \rangle] + \frac{L_{\max}}{2}\gamma_k^2 \mathbb{E}[\|\hat{g}_k\|_2^2]$$

$$= f(x_k) - \gamma_k \|g_k\|_2^2 + \frac{L_{\max}}{2}\gamma_k^2 \mathbb{E}[\|\hat{g}_k\|_2^2].$$

Taking full expectation, using variance bound assumption, we have

$$\mathbb{E}[f(x_{k+1})] - \mathbb{E}[f(x_k)] \leq -\gamma_k \mathbb{E}\|g_k\|_2^2 + \frac{L_{\max}}{2}\gamma_k^2 C$$

Therefore

$$\gamma_k \mathbb{E}\|g_k\|_2^2 \leq \mathbb{E}[f(x_k)] - \mathbb{E}[f(x_{k+1})] + \frac{C L_{\max}}{2}\gamma_k^2$$

Summing $k = 0, 1, \ldots, K-1$ gives

$$\sum_{k=0}^{K-1} \gamma_k \mathbb{E}\|g_k\|_2^2 \le (f(x_0) - f^*) + \frac{CL_{\max}}{2} \sum_{k=0}^{K-1} \gamma_k^2.$$

Now, in case of decreasing step sizes $\gamma_k = \gamma_0/\sqrt{k+1}$

$$\min_{0 \le k < K} \mathbb{E}\|g_k\|_2^2 \le \sum_{k=0}^{K-1} \frac{\gamma_0}{\sqrt{k+1}} \mathbb{E}\|g_k\|_2^2 \bigg/ \sum_{k=0}^{K-1} \frac{\gamma_0}{\sqrt{k+1}}$$

$$\le \frac{1}{\sqrt{K}} \left[ \frac{f(x_0) - f^*}{\gamma_0} + \frac{CL_{\max}}{2} \gamma_0 \sum_{k=0}^{K-1} \frac{1}{k+1} \right]$$

$$\le \frac{1}{\sqrt{K}} \left[ \frac{f(x_0) - f^*}{\gamma_0} + \gamma_0 CL_{\max} + \frac{\gamma_0 CL_{\max}}{2} \log K \right]$$

$$= \frac{1}{\sqrt{K}} \left[ \frac{f(x_0) - f^*}{\gamma_0} + \gamma_0 CL_{\max} \right] + \frac{\gamma_0 CL_{\max}}{2} \frac{\log K}{\sqrt{K}}.$$

where again we have used inequalities (22). In the case of constant step size $\gamma_k = \gamma$

$$\frac{1}{K} \sum_{k=0}^{K-1} \mathbb{E}\|g_k\|_2^2 \le \frac{1}{\gamma K} \left[ (f(x_0) - f^*) + \frac{CL_{\max}}{2} \gamma^2 K \right] = \frac{f(x_0) - f^*}{\gamma K} + \frac{CL_{\max}}{2} \gamma.$$

$\square$

# D   RECOVERING THEOREM 1 IN (BERNSTEIN ET AL., 2019) FROM THEOREM 1

To recover Theorem 1 in (Bernstein et al., 2019), first note that choosing a particular step size $\gamma$ in (7) yields

$$\frac{1}{K} \sum_{k=0}^{K-1} \mathbb{E}\|g_k\|_\rho \le \sqrt{\frac{2d\bar{L}(f(x_0) - f^*)}{K}}, \quad \text{with} \quad \gamma = \sqrt{\frac{2(f(x_0) - f^*)}{d\bar{L}K}}. \tag{31}$$

Then, due to Lemma 1, under unbiasedness and unimodal symmetric noise assumption, we can lower bound general $\rho$-norm by mixed $l^{1,2}$ norm. Finally we further lower bound our $l^{1,2}$ norm to obtain *the mixed norm* used in Theorem 1 of Bernstein et al. (2019): let $H_k = \{1 \le i \le d \colon \sigma_i < \sqrt{3}/2|g_{k,i}|\}$

$$5\sqrt{\frac{d\bar{L}(f(x_0) - f^*)}{K}} \ge \frac{5}{\sqrt{2}} \frac{1}{K} \sum_{k=0}^{K-1} \mathbb{E}\|g_k\|_\rho$$

$$\ge \frac{5}{\sqrt{2}} \frac{1}{K} \sum_{k=0}^{K-1} \mathbb{E}\|g_k\|_{l^{1,2}} = \frac{5}{\sqrt{2}} \frac{1}{K} \sum_{k=0}^{K-1} \left[ \sum_{i=1}^{d} \frac{g_i^2}{|g_i| + \sqrt{3}\sigma_i} \right]$$

$$\ge \frac{5}{\sqrt{2}} \frac{1}{K} \sum_{k=0}^{K-1} \mathbb{E} \left[ \frac{2}{5} \sum_{i \in H_k} |g_{k,i}| + \frac{\sqrt{3}}{5} \sum_{i \notin H_k} \frac{g_{k,i}^2}{\sigma_i} \right]$$

$$\ge \frac{1}{K} \sum_{k=0}^{K-1} \mathbb{E} \left[ \sum_{i \in H_k} |g_{k,i}| + \sum_{i \notin H_k} \frac{g_{k,i}^2}{\sigma_i} \right].$$

# E   STOCHASTIC SIGNSGD

Our experiments and the counterexample show that signSGD might fail to converge in general. What we proved is that SPB assumption is roughly a necessary and sufficient for general convergence. There are several ways to overcome SPB assumption and make signSGD to work in general, e.g.

scaled version of signSGD with error feedback (Karimireddy et al., 2019). Here we to present a simple way of fixing this issue, which is more natural to signSGD. The issue with signSGD is that sign of stochastic gradient is biased, which also complicates the analysis.

We define stochastic sign operator $\widetilde{\text{sign}}$, which unlike the deterministic sign operator is unbiased with appropriate scaling factor.

**Definition 3** (Stochastic Sign). *Define the stochastic sign operator* $\widetilde{\text{sign}} : \mathbb{R}^d \to \mathbb{R}^d$ *as*

$$\left(\widetilde{\text{sign}}\,g\right)_i = \begin{cases} +1, & \text{with prob. } \frac{1}{2} + \frac{1}{2}\frac{g_i}{\|g\|_2} \\ -1, & \text{with prob. } \frac{1}{2} - \frac{1}{2}\frac{g_i}{\|g\|_2} \end{cases}, \quad 1 \le i \le d,$$

*and* $\widetilde{\text{sign}}\,\mathbf{0} = \mathbf{0}$ *with probability 1.*

Furthermore, we define stochastic compression operator $\mathcal{C} : \mathbb{R}^d \to \mathbb{R}^d$ as $\mathcal{C}(x) = \|x\|_2 \cdot \widetilde{\text{sign}}\,x$, which compresses $rd$ bits to $r + d$ bits ($r$ bits per one floating point number). Then for any unbiased estimator $\hat{g}$ we get

$$\mathbb{E}\left[\mathcal{C}(\hat{g})\right] = \mathbb{E}\left[\mathbb{E}[\mathcal{C}(\hat{g})\,|\,\hat{g}]\right] = \mathbb{E}\left[\|\hat{g}\|_2\left(\frac{1}{2} + \frac{1}{2}\frac{\hat{g}}{\|\hat{g}\|_2}\right) - \|\hat{g}\|_2\left(\frac{1}{2} - \frac{1}{2}\frac{\hat{g}}{\|\hat{g}\|_2}\right)\right] = \mathbb{E}[\hat{g}] = g,$$

$$\text{Var}\left[\mathcal{C}(\hat{g})\right] = \mathbb{E}\left[\|\mathcal{C}(\hat{g}) - \hat{g}\|_2^2\right] = \mathbb{E}\left[\|\mathcal{C}(\hat{g})\|_2^2\right] - \mathbb{E}\left[\|\hat{g}\|_2^2\right] = (d-1)\mathbb{E}\|\hat{g}\|_2^2.$$

Using this relations, any analysis for SGD can be repeated for stochastic signSGD giving the same convergence rate with less communication and with $(d - 1)$ times worse coefficients.

Another scaled version of signSGD investigated in Karimireddy et al. (2019) uses non-stochastic compression operator $\mathcal{C}' : \mathbb{R}^d \to \mathbb{R}^d$ defined as $\mathcal{C}'(x) = \frac{\|x\|_1}{d}\,\text{sign}\,x$. It is shown (see Karimireddy et al. (2019), Theorem II) to converge as

$$\frac{1}{K}\sum_{k=0}^{K-1}\mathbb{E}\|\nabla f(x_k)\|_2^2 \le \frac{2\left(f(x_0) - f^*\right)}{\gamma K} + \frac{\gamma L_{\max} C}{2} + 4d(d-1)\gamma^2 L_{\max}^2 C,$$

where the error of current gradient compression is stored to be used in the next step. On the other hand, adopting the analysis of Theorem 6 for the stochastic compression operator $\mathcal{C}$, we get a bound

$$\frac{1}{K}\sum_{k=0}^{K-1}\mathbb{E}\|\nabla f(x_k)\|_2^2 \le \frac{f(x_0) - f^*}{\gamma K} + \frac{\gamma L_{\max} C d}{2},$$

where no data needs to be stored. Furthermore, ignoring the factor 2 at the first term, later bound is better if $\gamma \ge 1/8dL_{\max}$.

