# OpenReview forum: "On Stochastic Sign Descent Methods"
_ICLR.cc/2020/Conference — Reject_

### Official Review · AnonReviewer2 · 2019-10-23
**Official Blind Review #2**

**Rating:** 3

**Review:**

This paper performs a general analysis of sign-based methods for non-convex optimization. They define a new norm-like function depending on the success probabilities. Using this new norm-like function and under an assumption, they prove exponentially variance reduction properties in both directions and small mini-batch sizes.

I am not convinced about assumption 1, which plays the key role of the proof. It assumes that success probabilities are always large or equal to 1/2.

How can we guarantee this property hold for an algorithm? I suggest the authors provide some real learning examples, under which it will satisfy the condition.  I may revise my rating according to this.


**Experience Assessment:**

I have read many papers in this area.

**Review Assessment: Checking Correctness Of Derivations And Theory:**

I assessed the sensibility of the derivations and theory.

**Review Assessment: Checking Correctness Of Experiments:**

I assessed the sensibility of the experiments.

**Review Assessment: Thoroughness In Paper Reading:**

I made a quick assessment of this paper.

---

> ### Author Response · Authors · 2019-11-10
> **Thank you for the review!**
>
> We provided 3 different setups, where assumption 1 is satisfied:
> 1) Unimodal and symmetric noise setup (Lemma 1). As noted in (Bernstein et al., 2018), it is backed up by central limit theorem when training neural networks.
> 2) Strong growth condition with fixed mini-batch size (updated Lemma 2). This setup corresponds to over-parameterized deep network learning, where the model can fit the training data completely.
> 3) Adaptive mini-batch size setup (Lemma 3), which guarantees converge merely by choosing appropriate mini-batch size.
>
> Note that, while sign matching SPB assumption is quite intuitive in sign based methods, it is not assumed or somehow claimed that it holds automatically for simple problems. Even in one dimensional regression problem SPB assumption might fail, as much as signSGD might fail to converge. Furthermore, the SPB assumption describes the convergence of sign descent methods, which is known to be problematic (see e.g. (Balles & Hennig, 2018), section 6.2 Results).

---

### Official Review · AnonReviewer1 · 2019-10-23
**Official Blind Review #1**

**Rating:** 3

**Review:**



This paper focuses on signSGD with the aim of improving theoretical understanding of the method. The main contribution of the paper is to identify a condition SPB (success probability bounds), which is necessary for convergence of signSGD and study its connections with the other conditions known in the literature for signSGD analysis. One important point here is that the norm in which the authors show convergence now depends on SPB, meaning that the probabilities in SPB are used to define the norm-like function they use in the theorems.

This paper is well-written and nicely structured and I like the relationships of SPB with other conditions. However, I have some concerns on the generality of SPB that I will detail below.

- First of all, Lemma 2 is not clear to me at all. The authors say that the variance is bounded by a constant (0 \leq c_i < 1/sqrt{2}) multiplied by the true gradient norm and then they show that this assumption implies SPB. I do not know how restrictive this condition is. For example, what happens when all elements of true gradient is close to zero, I don’t know if it is reasonable to assume the noise to be small for this case. I cannot make the connection of this assumption and the classical bounded variance assumption (E((\hat{g_i}-g_i)^2)\leq\sigma_i). I can believe the result of Lemma 3 with specific constants $c_i$ as given, but I feel that it is then much stronger than standard bounded variance assumption. Because it would be asking the variance to be smaller than some specific constant.

- Related to first point, I did not understand the remark in the first footnote of page 2. The authors argue that SPB is weaker than bounded variance assumption. But at the same time, it is known that bounded variance assumption is not enough to make signSGD work, with counterexamples given in Karimireddy et. al. 2019. So, it is quite weird that an assumption weaker than bounded variance (for which signSGD provably does not convergence) makes signSGD converge. So I think it is more natural for SPB to be stronger than bounded variance, because it is enough to make signSGD work. The only proof in the paper that would support this claim is Lemma 2, as I discussed above is stronger than standard variance bound. I hope that authors can clarify this point.

- After Theorem 1, the authors compare their result with Bernstein et. al. 2019 and mention that Bernstein et. al. needs to use mini-batches depending on $K$ where $K$ is the iteration and unimodal symmetric noise assumption. But when I check Bernstein et. al. 2019, I see that these are different cases. Specifically, Theorem 1 in Bernstein et. al. 2019, uses mini-batch size 1 under unimodal symmetric noise assumption. The case where they would use mini-batches of size $K$ is in Section 3.3 of Bernstein et. al. 2019 where they *drop* unimodal symmetric noise assumption. So, I would suggest the authors to be more exact on this comparison because it is confusing. In fact, in Section 3.3 of Bernstein et. al. 2019, the authors also identify SPB as it is implied by unimodal symmetric noise assumption. It is the paragraph under Lemma 1 in Bernstein et. al. 2019.

- My other concern is the comparison with Karimireddy et. al. 2019 both in theory and practice. Karimireddy et. al. 2019 modifies signSGD and under unbiased stochastic gradients and bounded variance assumption, obtains similar guarantees as this paper. I am aware that this paper does not assume unbiasedness, but like I said before, I do not know how SPB compares to variance bound. So, I see Karimireddy et. al. 2019 and this paper as similar results, so I would want to see some practical comparison as well. In Appendix E, the authors mention that Karimireddy et. al. 2019 has storage need but I think that need is negligible since they only need to store one more vector.

- A side-effect of SPB is that now the convergence results are given in $\rho$-norm where $\rho$ is determined by SPB. I understand why this is needed from the proof of Theorem 1, and its implications in the theorem, but given that Karimireddy et. al. 2019’s result is given in l_2 norm which is easier to interpret, I think more comparison is needed.

- Lastly, I like the fact that SPB is implied by the previous assumption in Bernstein et. al. 2019, namely unimodal symmetric noise, I am not convinced that SPB is much weaker than this assumption. The authors mention in several places in the paper that their assumption is very weak, but looking at Lemma 1, Lemma 2 and Lemma 3: Lemma 1 and Lemma 3 are the already known cases where signSGD works, and Lemma 2 is a new case where signSGD works but as I explained before, it is not clear to me how restrictive this assumption is. Therefore, I am rather unsure if this generalization of SPB is practically significant.

Minor comments:
- page 2, Table 1: I think it would be useful to add the results of Karimireddy et. al. 2019.
- page 2, Table 1 and footnote 1: Footnote sign is given for the bounded stochastic gradient assumption but the explanation in the footnote text talks about the bounded variance assumption. Of course bounded stochastic gradient implies bounded variance, but this should be clarified. In addition, the footnote text is not clear to me, could the authors either point out to some references or give a proof?
- page 2, Adaptive methods paragraph: The end of the paragraph says that signSGD is similar to Adam so, studies on signSGD *can* improve the understanding of Adam. I would be happy if the authors are more exact about this, such as when signSGD is equivalent to Adam etc.
- page 3 discussion after Assumption 1: I do not understand the sentence starting with ‘Moreover, we argue that’. Can the authors give more details on why it is reasonable?
- page 4 Lemma 2: I think the authors should include the definition of variance in the paper. Since the assumption in this Lemma is rather non-standard, I think it makes sense to be as exact as possible.
page 21, Appendix E: It is written that ‘SPB is roughly a necessary and sufficient condition’. I could not understand what *roughly* means in this sentence. From what I have read, the authors have a counter-example showing without SPB, signSGD does not work and with SPB, it works, so I could not understand why it is written roughly here.

Overall, I like the generalization of SPB, but as I detail above, I am not sure how significant the generalization is compared to other results and more specifically how it compares to standard bounded variance (which I believe is weaker than Lemma 2). Therefore, I remain not convinced about the impact of this generalization hence the results. In addition, I would have liked to see more comparisons (both theoretical and practical) with Karimireddy et. al. 2019.

**Experience Assessment:**

I have published one or two papers in this area.

**Review Assessment: Checking Correctness Of Derivations And Theory:**

I carefully checked the derivations and theory.

**Review Assessment: Checking Correctness Of Experiments:**

I assessed the sensibility of the experiments.

**Review Assessment: Thoroughness In Paper Reading:**

I read the paper thoroughly.

---

> ### Author Response · Authors · 2019-11-10
> **Thank you for the detailed and constructive review! We have updated the paper and addressed the concerns you raised.**
>
> 1. The assumption of Lemma 2 (with an adjustment of fixed mini-batch size) can be seen as a Strong Growth Condition (SGC), sigma_i^2 \le C g_i^2, considered in the literature, e.g. (Vaswani et al., 2019; Schmidt and Le Roux, 2013). Under SGC optimal convergence rates for constant step-size SGD was obtained for convex, strongly convex and non-convex settings. Clearly, SGC a strong assumption, particularly it implies that all stochastic gradients are 0 at the stationary points. To be more precise, the assumption of Lemma 2 can be replaced by SGC with some constant C and requiring mini-batch size tau>2C. Under these assumptions, Lemma 2 yields lower bounds for success probabilities rho_i \ge (1 - C/tau) > 1/2 and Theorem 1 gives the rates (6),(7) with respect to l_1 norm instead of rho-norm. *We updated the Lemma 2*
> 2. With footnote of page 2 we argue slightly different claim. We do not argue that SPB is weaker than bounded variance assumption in a usual sense, but rather in the sense of differential entropy. In fact, these two assumptions are incomparable in the direct sense: neither SPB implies bounded variance assumption nor the other way around. We claim that under bounded variance assumption, the level of uncertainty of stochastic gradient is limited, while under the SPB assumption the information entropy of stochasticity could be infinite.
> 3. Again, we claim slightly different statement if you want to be more exact. We mention that convergence results in (Bernstein et al., 2018; 2019) use mini-batches *and/or* step-sizes dependent on K. Theorem 1 in (Bernstein et al., 2019) uses batch size 1, *but* step-size dependent on K. Other results, including the results for distributed setting, use mini-batch size *and* step size dependent on K. None of the results is free from K. By all means, we do not undermine the results in (Bernstein et al., 2018; 2019). In fact, these two papers were starting point for us and motivation to improve them.
> 4. To alleviate your concern for the comparison with (Karimireddy et. al. 2019), we point a couple of reasons of why we consider these results distinct: 1) First of all, the stochastic estimator in (Karimireddy et. al. 2019) is not just the signed vector, but it is scaled by l_1 norm of the gradient. Without that scaling factor, the results do not hold and cannot be applied to unscaled signSGD, which we consider. 2) Another difference is the incorporation of error-feedback mechanism, which uses unbiasedness of stochastic gradient and needs to store one more vector locally. Furthermore, we did compare ER-signSGD method of (Karimireddy et. al. 2019) with another scaled version of signSGD introduced in Appendix E, where we added a scaling factor as ER-signSGD does and we introduced extra stochasticity into sign vector instead of error-feedback.
> 5. It seems a fair trade-off between simplicity and generalization. SPB assumption as well as the notion of rho-norm are very general and convergence rates are hard to interpret. However, in special cases rho-norm can be l_1 norm (with strong growth condition) or mixed l_1-l_2 norm (with unimodal and symmetric noise assumption) which are easier to interpret.
> 6. To convince the weakness of SPB, for instance, with respect to unimodal symmetric noise assumption, consider the PDF(probability density function) of stochastic gradient. SPB requires the noise on the one side of axis, where the true gradient is, to concentrate more than the other side. In other words, median and true gradient must have the same sign, but the difference could be anything.Clearly, SPB allows multimodality and asymmetry for the noise. Lastly, while the case of Lemma 3 is known, the lower bound for the mini-batch size is adaptive, which is a novelty.
>
> Respond to minor comments
> 7. see respond 4.
> 8. We clarified footnote 1 and added some reference.
> 9. Mentioning that signSGD is similar to Adam, we also added a reference to (Balles & Henning, 2018) where the actual connection was made.
> 10. For each component, the sign of stochastic gradient has two possible values: +/- the sign of true gradient. It is easy to see that the sign of true gradient is a descent direction, while the other one is ascent direction. As we are aiming to solve a minimization problem, it is reasonable to follow the descent direction more often than the ascent one.
> 11. We updated Lemma 2, replacing current variance bound by strong growth condition, where the classical definition of variance is used.
> 12. SPB is not necessary and sufficient condition in the absolute sense, since one can (keeping unbiasedness) modify the stochastic gradient in the counterexample so that SPB does not hold but signSGD converges.
>
> It was well noted in (Balles & Hennig, 2018) that the usefulness of sign based methods is problem dependent. One should view the SPB condition as a criteria to a problem where sign based methods are useful. Moreover, taking account the counterexample, this criteria is not easily relaxable.

---

> > ### Comment · AnonReviewer1 · 2019-11-15
> > **Response to authors**
> >
> > I thank the authors for their response and I provide further remarks below:
> >
> > 1. Lemma 2 is more clear for me now, when the authors make the connection with SGC. I also agree that it is a very strong assumption under which people can actually prove very strong results for SGD. However, what is not clear to me is that under this assumption, $\tau$ depends on the constant $C$ of SGC. I think it would be clearer if the authors discuss how to estimate this constant.
> >
> > 2. Thank you for the clarification. However, the implication of this discussion in practice is still not very clear.
> >
> > 3. I think this part needs to be stated very clearly in the paper. To me, having a mini batch size depending on $K$ and step size depending on $K$ are very different cases which have to be treated separately.
> >
> > 4. I understand the difference but the scaling is done locally, during compression. Is there a drawback of scaling that I am missing? Second, I think storing one more vector is not a very big problem in practice. I think it would help to compare with this alternative approach in practice as well.
> >
> > 10. I accept this explanation as an intuition of course, but what I wanted to ask is that is this something the authors can show rigorously?
> >
> > The significance of the characterization of SPB, given that it is also mentioned in the paragraph under Lemma 1 in Bernstein et. al. 2019 (that I also pointed out in my original review), is still not enough for me. I think that the authors need more motivating cases in addition to Lemma 1, 2 and 3 which are either known (using big minibatch sizes or unimodal symmetric noise) or are very strong assumptions (SGC).

---

### Official Review · AnonReviewer3 · 2019-10-28
**Official Blind Review #3**

**Rating:** 6

**Review:**

The paper presents an improved analysis of the signSGD gradient estimator. The authors propose to relax the requirements on the gradient estimator in Bernstein (2019). The only requirement imposed on the gradient is that it should have the correct sign with probability greater than 1/2. In particular this approach allows the gradient estimate to be biased as opposed to Bernstein (2019) which requires unbiased gradients. The authors also show this condition to be necessary by a small counterexample.

In my view the paper presents a relatively minor but still interesting extension of the work in Bernstein (2019). The main problem is that the relaxation is not well motivated in terms of scenarios where this might be applicable. Experimental validation is also very weak.

It is claimed in the experiment section that the stochastic gradient of the Rosenbrock function g(x) = \del f_i(x) + eps, where eps is a 0-mean Gaussian and i is uniform random is biased. This seems incorrect to me and the gradient estimate should be unbiased when the expectation is taken over the randomness in i and eps.

A key claim of the paper is the ability to use biased gradient estimates. Experimental validation of this (in light of the above) is completely missing.

The experiments that are presented on MNIST are very general and not very closely connected to the specific claims of the paper. The only real conclusion drawn is that larger batch sizes improve convergence.

I think the paper needs better targeted experiments. They need to show covergence in a case where the conditions in Berstein (2019) do not hold.

How are the properties of the \rho norm related to the observations on l_1 norm for high and l_2 norm for low SNR components in Bernstein (2019)? If they are related this should be referenced.


**Experience Assessment:**

I have read many papers in this area.

**Review Assessment: Checking Correctness Of Derivations And Theory:**

I assessed the sensibility of the derivations and theory.

**Review Assessment: Checking Correctness Of Experiments:**

I carefully checked the experiments.

**Review Assessment: Thoroughness In Paper Reading:**

I read the paper at least twice and used my best judgement in assessing the paper.

---

> ### Author Response · Authors · 2019-11-10
> **Thank you for the review and suggestion for the experiments!**
>
> We motivated the relaxation by comparing it with 4 different conditions used in the literature: 1) unimodal symmetric noise assumption. 2) Strong growth condition with fixed mini-batch size. 3) Adaptive mini-batch size. 4) Bounded variance assumption in the sense of uncertainty.
>
> In fact the stochastic gradient of the Rosenbrock function considered in the paper is biased as expectation gives 1/(d-1) times the function itself.
> *We will add better targeted experiments.*
>
> The observation on low/high SNR components in (Bernstein et al., 2019) were done under the unimodal symmetric noise assumption. Similar observation is done in our paper for rho_norm and the second half of section 3 (including figure 1) is devoted just to that.
> *We added an explicit reference to (Bernstein et al., 2019) at the end of section 3.*

---

### Decision · Program_Chairs · 2019-12-19

**Decision:**

Reject

**Comment:**

This paper proposes an analysis of signSGD in some special cases. SignGD has been shown to be of interest, whether because of its similarity to Adam or in quasi-convex settings.

The complaint shared by reviewers was the strength of the conditions. SGC is really strong, I have yet to see increasing mini-batch sizes to be used in practice (although there are quite a  few papers mentioning this technique to get a convergence rate) and the strength of the other two are harder to assess. With that said, the improvement compared to existing work such as Karimireddy et. al. 2019 is unclear.

I encourage the authors to address the comment of the reviewers and to submit an improved version to a later, or perhaps to a more theoretical, convergence.